# NEYMAN-PEARSON CLASSIFICATION UNDER BOTH NULL AND ALTERNATIVE DISTRIBUTIONS SHIFT

**Mohammadreza M. Kalan**
Univ Rennes, Ensai, CNRS,
CREST–UMR 9194, F-35000 Rennes, France
mohammadreza.kalan@ensai.fr

**Yuyang Deng**
Columbia University, Department of Statistics
yd2824@columbia.edu

**Eitan J. Neugut**
Columbia University, Department of Statistics
eitan.neugut@columbia.edu

**Samory Kpotufe**
Columbia University, Department of Statistics
samory@columbia.edu

## ABSTRACT

We consider the problem of transfer learning in Neyman–Pearson classification, where the objective is to minimize the error w.r.t. a distribution $\mu_1$, subject to the constraint that the error w.r.t. a distribution $\mu_0$ remains below a prescribed threshold. While transfer learning has been extensively studied in traditional classification, transfer learning in imbalanced classification such as Neyman–Pearson classification has received much less attention. This setting poses unique challenges, as both types of errors must be simultaneously controlled. Existing works address only the case of distribution shift in $\mu_1$, whereas in many practical scenarios shifts may occur in both $\mu_0$ and $\mu_1$. We derive an adaptive procedure that not only guarantees improved Type-I and Type-II errors when the source is informative, but also automatically adapt to situations where the source is uninformative, thereby avoiding negative transfer. In addition to such statistical guarantees, the procedures is efficient, as shown via complementary computational guarantees.

## 1 INTRODUCTION

In many applications, the objective is to learn a decision rule from data that separates two classes with distributions $\mu_0$ and $\mu_1$, whose sample sizes are often imbalanced. These applications include disease diagnosis (Myszczynska et al., 2020; Bourzac, 2014), malware detection in cybersecurity (Alamro et al., 2023; Kumar & Lim, 2019), and climate science, such as heavy rain detection (Folino et al., 2023; Frame et al., 2022). In such settings, the learner must control the error w.r.t. both classes. In traditional classification, the goal is to minimize the total loss, and under class imbalance this often leads to a classifier with a large error on the rare class. A natural framework to control errors with respect to both classes is Neyman–Pearson (NP) classification (Cannon et al., 2002; Scott & Nowak, 2005), where the objective is to minimize the error with respect to $\mu_1$, subject to the constraint that the error with respect to $\mu_0$ does not exceed a pre-specified threshold $\alpha$, usually a small value (for example, 5% Tong et al. (2018)). In NP classification, even when one class is rare relative to the other, the number of available samples from both classes may still be limited. To mitigate data scarcity in the primary classification task (referred to as the *target*), where samples from either class may be limited, one can make use of additional datasets (referred to as the *source*). However, while transfer learning has been extensively studied in traditional classification, how to effectively leverage source data in the NP classification has received far less attention.

In this work, we propose a provable procedure that effectively leverages the source under potential distribution shifts in both $\mu_0$ and $\mu_1$ to improve both Type-I error, i.e., the error w.r.t. $\mu_0$, and Type-II error, i.e., the error w.r.t. $\mu_1$. Prior theoretical works (Kalan et al., 2025; Kalan & Kpotufe, 2024b) address only the special case where the source shares the same $\mu_0$ with the target and the shift occurs solely in $\mu_1$. However, in many applications, shifts may occur in both distributions, raising the challenge of how to exploit the source to improve both types of errors simultaneously.

A key difficulty in the presence of a shift in $\mu_0$ is that a classifier satisfying the source constraint, i.e., achieving source Type-I error below $\alpha$, does not necessarily satisfy the target constraint. Especially when only a few target samples are available from $\mu_0$, the set of classifiers that meet the empirical $\alpha$ constraint on these samples may still substantially exceed the true population risk w.r.t. $\mu_0$. Moreover, one cannot straightforwardly use the source to rule out classifiers that result in large population error w.r.t. $\mu_0$ in target, since the correspondence between the source constraint and the target constraint is unknown. Thus, the central challenge is how to leverage source samples to control the target Type-I error while simultaneously improving the target Type-II error—an issue that does not arise in the setting considered by prior works (Kalan et al., 2025; Kalan & Kpotufe, 2024b).

The procedure proposed in this work adaptively leverages the source without requiring any prior knowledge about its relatedness to the target. When the source is informative, it improves both types of errors on the target; when the source is uninformative, it matches the performance of using only target data, thereby avoiding negative transfer. Moreover, the generalization error bound derived in this work recovers the bounds established in prior works (Kalan et al., 2025; Kalan & Kpotufe, 2024b) in the special case where the source and target share the same distribution $\mu_0$. The proposed procedure consists of two stages. In the first stage, it utilizes source class-0 samples to determine an effective threshold $\hat{\alpha}_S$ that aligns the source Type-I constraint with the target constraint. This step rules out classifiers that would exceed the target Type-I constraint $\alpha$, while retaining those capable of achieving low Type-II error on the target. In the second stage, it leverages source class-1 samples to further reduce the Type-II error on the target while ensuring that the Type-I error remains below the prescribed threshold.

In addition to a statistical guarantee for the proposed procedure, we establish a computational guarantee by reformulating the learning procedure as a constrained optimization problem. Specifically, we design a computationally feasible procedure that reduces the learning procedure to a sequence of convex programs and leverages a stochastic convex optimization solver as an algorithmic component. We show that this optimization procedure results in a model that achieves the statistical guarantee in polynomial time.

## 2 RELATED WORK

Transfer learning has been extensively studied in traditional classification and regression (Li et al., 2022; Cai & Wei, 2021; Kpotufe & Martinet, 2021; Tripuraneni et al., 2020; Mousavi Kalan et al., 2020; Kalan et al., 2022; Ben-David et al., 2010; 2006; Mansour et al., 2009). A popular approach in vanilla classification is $\alpha$-ERM (Bu et al., 2022; Aminian et al., 2024), which assigns weights $\alpha$ and $1 - \alpha$ to the target and source empirical losses. Another widely used approach is fine-tuning (Vrbančič & Podgorelec, 2020), where a model trained on the source domain is further refined using target data. However, NP classification requires controlling the Type-I error, leading to a constrained problem that minimizes the Type-II error under a Type-I constraint. These unconstrained methods do not address this requirement.

A closely related work is Hanneke & Kpotufe (2019), which studies transferability across domains under distribution shift and develops an adaptive transfer learning approach. They introduce a notion of transfer distance that quantifies how performance on the source domain transfers to the target, establish minimax rates under this measure, and show that their procedure adapts to the unknown transfer distance in balanced classification. However, their setting does not address imbalanced problems, where controlling both Type-I and Type-II errors simultaneously is the main challenge.

The problem of NP classification when the underlying distributions are unknown except through samples was first formulated by Cannon et al. (2002); Scott & Nowak (2005), who considered empirical risk minimization with respect to one class while constraining the empirical error on the other below a pre-specified threshold. Rigollet & Tong (2011) studied the same problem under a surrogate convex loss. Tong (2013) studied a nonparametric NP classification framework and established rates of convergence for a plug-in approach based on estimating the class distributions. More recently, Kalan & Kpotufe (2024a) derived distribution-free minimax rates for NP classification and showed that, unlike traditional classification, the problem exhibits a dichotomy between fast and slow rates.

Related to this work, in the context of transfer learning in NP classification, Kalan & Kpotufe (2024b) derived minimax rates for 0-1 loss. Building on this, Kalan et al. (2025) introduced an

implementable transfer learning procedure for NP classification using a surrogate loss and derived upper bound guarantees. Both Kalan & Kpotufe (2024b); Kalan et al. (2025) focused on the restricted setting where the source and target share the class-0 distribution $\mu_0$ and the shift occurs only in $\mu_1$. Moreover, they established only statistical guarantees without addressing computational aspects. However, in this work, we consider the general setting where distribution shifts may occur in both $\mu_0$ and $\mu_1$. We propose an adaptive procedure with statistical guarantees that also recovers the special case in Kalan & Kpotufe (2024b); Kalan et al. (2025). On the computational side, we reformulate the learning problem within a two-stage convex programming framework and develop a concrete stochastic optimization procedure that outputs a model with the desired excess-risk guarantee with a bound on the gradient complexity of the procedure.

## 3   SETUP

Let $(\mathcal{X}, \Sigma)$ be a measurable space, and let $\mathcal{H}$ denote a hypothesis class of measurable functions $h : \mathcal{X} \to \mathbb{R}$. In the binary classification setting, each $h \in \mathcal{H}$ induces a decision rule of the form $\mathbb{1}\{h(x) \geq 0\}$, which assigns a sample $x$ to either class 0 or class 1. In this paper, we adopt the NP classification framework. Classes 0 and 1 are generated according to distributions $\mu_0$ and $\mu_1$, respectively, and the goal is to learn a function $\hat{h} \in \mathcal{H}$ from samples drawn from these distributions. The classifier $\hat{h}$ is required to keep the error w.r.t. class 0 (i.e., the Type-I error) below a pre-specified threshold $\alpha$, while minimizing the error w.r.t. class 1 (i.e., the Type-II error). We first introduce a loss function w.r.t. which the Type-I and Type-II errors are defined.

**Definition 1** (Surrogate Loss). *We call a function $\varphi : \mathbb{R} \to \mathbb{R}_+$ an L-Lipschitz surrogate loss whenever the following hold: $\varphi$ is monotone nondecreasing and normalized by $\varphi(0) = 1$; it satisfies the Lipschitz bound $|\varphi(x) - \varphi(y)| \leq L|x - y|$ for all $x, y \in \mathbb{R}$; and there exists some constant $C > 0$ such that, for every $h \in \mathcal{H}$ and $x \in \mathcal{X}$, we have $\max\{\varphi(h(x)), \varphi(-h(x))\} \leq C$.*

The boundedness requirement in Definition 1 is a standard assumption in learning theory, ensuring that the empirical loss uniformly concentrates around the population counterpart (Bartlett & Mendelson, 2002). This assumption is not restrictive in practice. Many commonly used surrogate losses (such as logistic or hinge losses) are clipped in implementations for numerical stability, which directly enforces boundedness. Moreover, boundedness naturally holds when the hypothesis class is norm-bounded, as is typical for linear models or neural networks with bounded parameters.

Next, we define Type-I and Type-II errors w.r.t. a surrogate loss.

**Definition 2.** *For a surrogate loss $\varphi$, the $\varphi$-Type-I and $\varphi$-Type-II errors of $h$ are defined as $R_{\varphi,\mu_0}(h) = \mathbb{E}_{\mu_0}[\varphi(h(X))]$ and $R_{\varphi,\mu_1}(h) = \mathbb{E}_{\mu_1}[\varphi(-h(X))]$, respectively.*

In particular, for the indicator loss $\varphi(z) = \mathbb{1}\{z \geq 0\}$, the errors $R_{\varphi,\mu_0}$ and $R_{\varphi,\mu_1}$ reduce to the standard Type-I and Type-II errors.

The NP classification problem w.r.t. a surrogate loss $\varphi$ and a pre-specified threshold $\alpha$ is defined as

$$\min_{h \in \mathcal{H}} R_{\varphi,\mu_1}(h) \quad \text{s.t.} \quad R_{\varphi,\mu_0}(h) \leq \alpha \tag{1}$$

and we denote by $h_\alpha^*$ a (not necessarily unique) solution to (1). Furthermore, note that when $\mathcal{H}$ contains all measurable functions from $\mathcal{X}$ to $\mathbb{R}$ and $\varphi$ is the indicator loss, the NP Lemma (Lehmann & Lehmann, 1986) characterizes the solution to (1) as $h_\alpha^* = 2\mathbb{1}\left\{\frac{p_1(x)}{p_0(x)} \geq \lambda\right\} - 1$, for a suitable $\lambda$, under mild regularity conditions, where $p_0$ and $p_1$ denote the class-conditional densities of classes.

In a practical setting, the learner does not have access to the distributions $\mu_0$ and $\mu_1$ except through observing i.i.d. samples from each class: $\{X_i^{(0)}\}_{i=1}^{n_0} \sim \mu_0$ and $\{X_i^{(1)}\}_{i=1}^{n_1} \sim \mu_1$. Then, the corresponding empirical $\varphi$-Type-I and $\varphi$-Type-II errors are

$$\widehat{R}_{\varphi,\mu_0}(h) := \frac{1}{n_0} \sum_{i=1}^{n_0} \varphi\big(h(X_i^{(0)})\big), \qquad \widehat{R}_{\varphi,\mu_1}(h) := \frac{1}{n_1} \sum_{i=1}^{n_1} \varphi\big(-h(X_i^{(1)})\big).$$

(Cannon et al., 2002; Scott & Nowak, 2005) showed that by the following constrained empirical risk minimization

$$\hat{h} = \arg\min_{h \in \mathcal{H}} \widehat{R}_{\varphi,\mu_1}(h) \quad \text{s.t.} \quad \widehat{R}_{\varphi,\mu_0}(h) \leq \alpha + \epsilon_0 \tag{2}$$

where $\epsilon_0 = \frac{\tilde{C}}{\sqrt{n_0}}$ for an appropriate constant $\tilde{C}$, one can obtain that $R_{\varphi,\mu_1}(\hat{h}) - R_{\varphi,\mu_1}(h^*_\alpha) \lesssim \frac{1}{\sqrt{n_1}}$ and $R_{\varphi,\mu_0}(\hat{h}) - \alpha \lesssim \frac{1}{\sqrt{n_0}}$.

## 3.1 TRANSFER LEARNING SETUP

We consider a transfer learning setting with source and target domains, where for each $D \in \{S, T\}$ the class 0 and class 1 distributions are denoted by $\mu_{0,D}$ and $\mu_{1,D}$. The NP classification problem w.r.t. a surrogate loss $\varphi$ and a pre-specified threshold $\alpha$ in each domain $D \in \{S, T\}$ is defined as

$$\min_{h \in \mathcal{H}} R_{\varphi,\mu_{1,D}}(h) \quad \text{s.t. } R_{\varphi,\mu_{0,D}}(h) \leq \alpha \tag{3}$$

and we denote by $h^*_{D,\alpha}$ a (not necessarily unique) solution to (3). Moreover, the learner has access to i.i.d. samples from each class in both domains: $\{X_i^{(0,D)}\}_{i=1}^{n_{0,D}} \sim \mu_{0,D}$ and $\{X_i^{(1,D)}\}_{i=1}^{n_{1,D}} \sim \mu_{1,D}$ for $D \in \{S, T\}$. Then, the corresponding empirical $\varphi$-Type-I and $\varphi$-Type-II errors for $D \in \{S, T\}$ are

$$\widehat{R}_{\varphi,\mu_{0,D}}(h) := \frac{1}{n_{0,D}} \sum_{i=1}^{n_{0,D}} \varphi\big(h(X_i^{(0,D)})\big), \qquad \widehat{R}_{\varphi,\mu_{1,D}}(h) := \frac{1}{n_{1,D}} \sum_{i=1}^{n_{1,D}} \varphi\big(-h(X_i^{(1,D)})\big).$$

The goal of the learner in a transfer learning setting is to learn a function $\hat{h} \in \mathcal{H}$ using $n_{0,S}, n_{1,S}, n_{0,T}, n_{1,T}$ samples from the source and target domains, such that it performs well on the *target* domain. Specifically, the learner's goal is to minimize the target $\varphi$-Type-II excess error

$$\mathcal{E}_{1,T}(\hat{h}) := \big[R_{\varphi,\mu_{1,T}}(\hat{h}) - R_{\varphi,\mu_{1,T}}(h^*_{T,\alpha})\big]_+, \qquad [u]_+ := \max\{0, u\},$$

subject to the $\varphi$-Type-I error constraint $R_{\varphi,\mu_{0,T}}(\hat{h}) \leq \alpha + \epsilon_{0,T}$, where $\epsilon_{0,T}$ is of order $n_{0,T}^{-1/2}$.

In this paper, we aim to develop an adaptive procedure that, without requiring any prior knowledge about the relatedness between source and target, effectively leverages both types of samples with two guarantees. First, regardless of whether the source is related to the target, it is as good as using only target samples and ignoring the source. Hence, it avoids negative transfer. Second, whenever the source is related to the target, it improves upon the performance of using only target samples.

Kalan et al. (2025); Kalan & Kpotufe (2024b) develop adaptive procedures for the special case where $\mu_{0,T} = \mu_{0,S}$. They stablish the bounds $R_{\varphi,\mu_{0,T}}(\hat{h}) - \alpha \lesssim \frac{1}{\sqrt{n_0}}$, where $n_0$ denotes the number of samples from $\mu_0 = \mu_{0,S} = \mu_{0,T}$, and

$$\mathcal{E}_{1,T}(\hat{h}) \lesssim \min\{\frac{1}{\sqrt{n_{1,T}}}, R_{\varphi,\mu_{1,T}}(h^*_{S,\alpha}) - R_{\varphi,\mu_{1,T}}(h^*_{T,\alpha}) + (\frac{1}{\sqrt{n_{1,S}}})^{1/\rho}\}$$

where $\rho$ is the transfer exponent (Hanneke & Kpotufe, 2019), which quantifies the relatedness between source and target.

**Challenges of transfer learning in a general setting:** When $\mu_{0,S} \neq \mu_{0,T}$ the feasible set in (3) under the target constraint need not coincide with its source counterpart. Therefore, a classifier that satisfies the source $\varphi$-Type-I at level $\alpha$ need not satisfy the target constraint. More precisely, let $\mathcal{H}_D(\alpha) := \{h \in \mathcal{H} : R_{\varphi,\mu_{0,D}}(h) \leq \alpha\}$ for $D \in \{S, T\}$. If $\mu_{0,S} \neq \mu_{0,T}$, then $h \in \mathcal{H}_S(\alpha)$ does not imply $h \in \mathcal{H}_T(\alpha)$.

Furthermore, since the learner only observes samples rather than having direct access to the distributions, the empirical set $\hat{\mathcal{H}}_T(\alpha) := \{h \in \mathcal{H} : \hat{R}_{\varphi,\mu_{0,T}}(h) \leq \alpha + \epsilon_{0,T}\}$ may yield a $\varphi$-Type-I error substantially larger than $\alpha$, especially when $n_{0,T}$ is small and consequently $\epsilon_{0,T}$ is large. In addition, one does not know in advance an appropriate value $\alpha' \in [0, 1]$ in the source domain such that exploiting $\hat{\mathcal{H}}_S(\alpha') = \{h \in \mathcal{H} : \hat{R}_{\varphi,\mu_{0,S}}(h) \leq \alpha' + \epsilon_{0,S}\}$, where $\epsilon_{0,S} = \frac{\tilde{C}}{\sqrt{n_{0,S}}}$, would lead to improved $\varphi$-Type-I performance in target. Choosing too small an $\alpha'$ in source may reduce the $\varphi$-Type-I error in target, but at the cost of a substantially larger $\varphi$-Type-II error in target.

These challenges motivate the need for an adaptive transfer learning procedure, which we develop in the next section.

## 4 ADAPTIVE TRANSFER LEARNING PROCEDURE

We start with defining the smallest $\alpha' \in [0, 1]$ such that the constrained set in the source, i.e. $\mathcal{H}_S(\alpha')$ contains the optimal function in the target as:

$$\alpha_S := \inf\{\alpha' : \mathcal{H}_S(\alpha') \cap T^*(\alpha) \neq \emptyset\} \tag{4}$$

where $T^*(\alpha) \subset \mathcal{H}$ denote the set of solutions of Target problem (3). Next, we define the set of functions whose empirical $\varphi$-Type-I error in the target satisfies the $\alpha$ constraint, and whose empirical $\varphi$-Type-II error in the target domain does not exceed that of the corresponding empirical risk minimizer. Specifically,

$$\hat{\mathcal{H}}_{\alpha,T}^* := \left\{ h \in \hat{\mathcal{H}}_T(\alpha) : \hat{R}_{\varphi,\mu_{1,T}}(h) \leq \hat{R}_{\varphi,\mu_{1,T}}(\hat{h}_{T,\alpha-\epsilon_{0,T}}^*) + 6\epsilon_{1,T} \right\}, \tag{5}$$

where $\epsilon_{1,T} = \frac{\tilde{C}}{\sqrt{n_{1,T}}}$ and $\hat{h}_{T,\alpha-\epsilon_{0,T}}^* := \underset{h \in \hat{\mathcal{H}}_T(\alpha-\epsilon_{0,T})}{\arg\min} \hat{R}_{\varphi,\mu_{1,T}}(h)$ . The population $\varphi$-Type-I error

of functions in $\hat{\mathcal{H}}_{\alpha,T}^*$ can substantially exceed $\alpha$ when the number of target samples from $\mu_{0,T}$, i.e., $n_{0,T}$, is small. To address this, we exploit the source domain by restricting it to a suitable subset, thereby retaining only functions with lower $\varphi$-Type-I error. Specifically, we first define

$$\hat{\alpha}_S := \inf \left\{ \alpha' \in [\alpha, 1] : \hat{\mathcal{H}}_S(\alpha') \cap \hat{\mathcal{H}}_{\alpha,T}^* \neq \emptyset \right\}. \tag{6}$$

and then introduce the restricted set $\hat{\mathcal{H}}' = \hat{\mathcal{H}}_S(\hat{\alpha}_S) \cap \hat{\mathcal{H}}_{\alpha,T}^*$. We will show formally later that the functions in $\hat{\mathcal{H}}'$ satisfy the required bound on the $\varphi$-Type-I error. At this stage, we need to further restrict $\hat{\mathcal{H}}'$ to achieve a small $\varphi$-Type-II error. For this purpose, we introduce the following sets for $D \in \{S, T\}$:

$$\hat{\mathcal{H}}_{1,D}' := \{h \in \hat{\mathcal{H}}' : \hat{R}_{\varphi,\mu_{1,D}}(h) \leq \hat{R}_{\varphi,\mu_{1,D}}^*(\hat{\mathcal{H}}') + 2\epsilon_{1,D}\} \tag{7}$$

where $\epsilon_{1,D} = \frac{\tilde{C}}{\sqrt{n_{1,D}}}$ and $\hat{R}_{\varphi,\mu_{1,D}}^*(\hat{\mathcal{H}}') = \min_{h \in \hat{\mathcal{H}}'} \hat{R}_{\varphi,\mu_{1,D}}(h)$. Now, equipped with these definitions,

we propose the following transfer learning procedure, which outputs a function $\hat{h} \in \mathcal{H}$.

---

If $\hat{\mathcal{H}}_{1,S}' \cap \hat{\mathcal{H}}_{1,T}' \neq \emptyset$, then choose $\hat{h} \in \hat{\mathcal{H}}_{1,S}' \cap \hat{\mathcal{H}}_{1,T}'$.

Otherwise, choose $\hat{h} = \underset{h \in \hat{\mathcal{H}}'}{\arg\min} \hat{R}_{\varphi,\mu_{1,T}}(h)$. $\tag{8}$

---

## 5 MAIN RESULTS

### 5.1 GENERALIZATION ERROR BOUNDS

We begin with the definition of Rademacher complexity, a capacity measure used to ensure uniform convergence guarantees between the empirical and population risks.

**Definition 3** (Rademacher Complexity (Bartlett & Mendelson, 2002)). *Let $X_1, \ldots, X_n$ be i.i.d. random variables drawn from a distribution $\mu$ on $\mathcal{X}$. Let $\sigma_1, \ldots, \sigma_n$ be independent Rademacher variables, i.e. random signs with $\Pr(\sigma_i = +1) = \Pr(\sigma_i = -1) = \frac{1}{2}$. The empirical Rademacher complexity of $\mathcal{H}$ is defined as $\hat{R}_n(\mathcal{H}) = \mathbb{E}_\sigma \left[ \sup_{h \in \mathcal{H}} \frac{1}{n} \sum_{i=1}^n \sigma_i h(X_i) \right]$ . The Rademacher complexity of $\mathcal{H}$ is then $R_n(\mathcal{H}) = \mathbb{E}_{X_1^n} \left[ \hat{R}_n(\mathcal{H}) \right]$ .*

**Assumption 1** (Class Complexity). *We assume that the complexity of the hypothesis class satisfies $R_n(\mathcal{H}) \leq \frac{B_{\mathcal{H}}}{\sqrt{n}}$ for some constant $B_{\mathcal{H}}$ capturing the complexity of $\mathcal{H}$.*

Note that when the input features are bounded, many hypothesis classes used in practice—such as linear models or neural networks with bounded parameters—satisfy Assumption 1 (Golowich et al., 2018). Furthermore, we make the following convexity assumption on the hypothesis class, which, together with the convexity of the loss, ensures that small deviations in the $\varphi$-Type-I error do not lead to large deviations in the $\varphi$-Type-II error. The bounds can also be derived without this assumption, though it yields tighter bounds.

**Assumption 2** (Class Convexity). *The class $\mathcal{H}$ is convex: given any two hypotheses $h_1, h_2 \in \mathcal{H}$ and any $\theta \in (0, 1)$, we have $\theta h_1 + (1 - \theta)h_2 \in \mathcal{H}$.*

Note that polynomial regression functions and majority votes over a set of basis functions are examples that satisfy Assumption 2. In contrast, a neural network class with a fixed architecture is generally not closed under convex combinations. Nevertheless, because the Rademacher complexity of a class matches that of its convex hull, the convex hull of a neural network class can be considered instead, which is convex (Kalan et al., 2025; Rigollet & Tong, 2011)

Next, we need a notion of distance between the source and target that translates the performance of a function $h \in \mathcal{H}$ on the source to its performance on the target. We first introduce a general notion of distance and then derive bounds on the generalization error in terms of this notion. This formulation is broad enough to also yield bounds expressed through the commonly used notion of transfer exponent (Hanneke & Kpotufe, 2019; Kalan & Kpotufe, 2024b).

Let us define $\mathcal{H}^*_{S,T,\alpha} := \underset{h \in \mathcal{H}_S(\alpha) \cap \mathcal{H}_T(\alpha)}{\arg\min} R_{\varphi, \mu_{1,S}}(h)$, which denotes the set of solutions minimizing the source $\varphi$-Type-II risk within the intersection of $\alpha$ feasible sets in the source and target. Among these solutions, let $h^*_{S,T,\alpha} := \arg\max_{h \in \mathcal{H}^*_{S,T,\alpha}} R_{\varphi, \mu_{1,T}}(h)$ denote the pivoting function serving as the reference for comparing errors across domains in the definition of transfer distance. We then define the excess error of any hypothesis $h \in \mathcal{H}$ w.r.t. this pivoting function, for $D \in \{S, T\}$, as

$$\mathcal{E}_{1,D}(h \mid h^*_{S,T,\alpha}) := R_{\varphi, \mu_{1,D}}(h) - R_{\varphi, \mu_{1,D}}(h^*_{S,T,\alpha}). \tag{9}$$

Using this, we define the transfer modulus, which translates the performance of a function $h \in \mathcal{H}$ from source to target:

**Definition 4** (Transfer Modulus). *For $\varepsilon \geq 0$, the transfer moduli for $\varphi$-Type-I and $\varphi$-Type-II errors are defined as follows*

$$\phi_1^{S \to T}(\varepsilon) := \sup \left\{ \mathcal{E}_{1,T}(h \mid h^*_{S,T,\alpha}) : h \in \mathcal{H}, \mathcal{E}_{1,S}(h \mid h^*_{S,T,\alpha}) \leq \varepsilon \right\},$$

$$\phi_0^{S \to T}(\varepsilon) := \sup \left\{ R_{\varphi, \mu_{0,T}}(h) : h \in \mathcal{H}, R_{\varphi, \mu_{0,S}}(h) \leq \varepsilon \right\}.$$

Now, equipped with these definitions, we state the following theorem, which provides upper bounds on the generalization error of the proposed transfer learning procedure in Section 4.

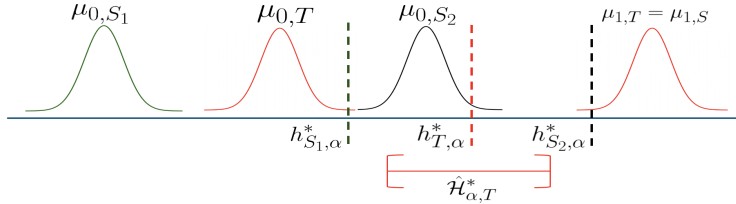

Figure 1: In this figure, we consider one target and two sources that all share the distribution $\mu_1$, while $\mu_0$ differs across them. All distributions are Gaussian with the same variance but different means. The optimal NP classifiers are denoted by $h^*_{T,\alpha}$, $h^*_{S_1,\alpha}$, and $h^*_{S_2,\alpha}$. Moreover, we assume that there are sufficiently many target samples such that $\hat{\mathcal{H}}^*_{\alpha,T}$ does not intersect with $h^*_{S_1,\alpha}$ or $h^*_{S_2,\alpha}$, which implies that $\alpha_{S_2} > \alpha_{S_1} = \alpha$

**Theorem 1.** *Suppose that the hypothesis class $\mathcal{H}$ satisfies Assumptions 1 and 2. Moreover, let $\delta > 0$ and $\epsilon_{i,D} = \frac{\tilde{C}}{\sqrt{n_{i,D}}}$ for $i \in \{0, 1\}$ and $D \in \{S, T\}$, where $\tilde{C} = 8B_{\mathcal{H}}L + 2C\sqrt{2 \log\left(\frac{2}{\delta}\right)}$. Assume further that $n_{0,T} \geq n_{1,T}$, that $\mathcal{H}_T(\alpha/8)$ is nonempty, and that $n_{0,T}$ is large enough so that $\epsilon_{0,T} \leq \frac{7\alpha}{16}$. Furthermore, let $\varphi$ be a convex surrogate loss function, and let $\hat{h}$ denote the hypothesis returned by the procedure in Section 4. Then, with probability at least $1 - 4\delta$, we have:*

$$\mathcal{E}_{1,T}(\hat{h}) \leq c \cdot \min \left\{ \epsilon_{1,T}, R_{\varphi, \mu_{1,T}}(h^*_{S,T,\alpha}) - R_{\varphi, \mu_{1,T}}(h^*_{T,\alpha}) + \phi_1^{S \to T}(4\epsilon_{1,S}) \right\}$$

$$R_{\varphi, \mu_{0,T}}(\hat{h}) \leq \begin{cases} \min\{\alpha + 2\epsilon_{0,T}, \phi_0^{S \to T}(\alpha + 2\epsilon_{0,S})\}, & \text{if } \alpha \geq \alpha_S, \\ \min\{\alpha + 2\epsilon_{0,T}, \phi_0^{S \to T}(\hat{\alpha}_S + 2\epsilon_{0,S})\}, & \text{if } \alpha < \alpha_S. \end{cases}$$

*where $c$ is a universal constant and $\hat{\alpha}_S$ is the empirical threshold defined in (6). In particular, if $\alpha < \alpha_S$, then $\hat{\alpha}_S \leq \alpha_S$.*

Note that the assumption $n_{0,T} \geq n_{1,T}$ typically holds in practice, since in the NP setting class 1 usually corresponds to the rare class, in contrast to class 0. Furthermore, without this assumption, only the bound for the $\varphi$-Type-II error would change, where the term $c \cdot \epsilon_{1,T}$ would be replaced by $c \cdot (\epsilon_{0,T} + \epsilon_{1,T})$.

**Remark 1.** *In the special case where $\mu_{0,S} = \mu_{0,T}$, Theorem 1 recovers the bounds in Kalan et al. (2025); Kalan & Kpotufe (2024b). In this setting, we have $\alpha_S \leq \alpha$ by (4), and $\phi_0^{S \to T}$ reduces to the identity function. Consequently, the bound for the $\varphi$-Type-I error simplifies to $\alpha + \epsilon_0$, where $\epsilon_0 = \epsilon_{0,S} = \epsilon_{0,T}$. Regarding the $\varphi$-Type-II excess error bound, in this case we have $h_{S,T,\alpha}^* = h_{S,\alpha}^*$. Furthermore, in the case where source and target distributions are identical, i.e., $\mu_{i,S} = \mu_{i,T}$ for $i \in \{0,1\}$, as $n_{0,S}, n_{1,S} \to \infty$, the $\varphi$-Type-II excess error converges to 0, while the upper bound on the $\varphi$-Type-I error converges to $\alpha$.*

**Remark 2.** *Theorem 1 guarantees the adaptivity of the proposed transfer learning procedure without requiring any prior knowledge of the relatedness between source and target. It recovers the performance of using only target data, thereby avoiding negative transfer when the source is uninformative. Moreover, whenever the source is related and informative, the procedure leverages it to achieve better performance compared to using only the target, in terms of both $\varphi$-Type-I and $\varphi$-Type-II errors.*

**Remark 3.** *In Theorem 1, the source effect is captured by the functions $\phi_1^{S \to T}$ and $\phi_0^{S \to T}$. In contrast, Kalan et al. (2025); Kalan & Kpotufe (2024b) use the notion of transfer exponent (Hanneke & Kpotufe, 2019) to characterize the source effect. In Appendix A, we show that Theorem 1 can also be expressed in terms of the transfer exponent.*

**Remark 4.** *Figure 1 illustrates the effect of $\alpha_S$ on reducing the target Type-I error. In this example, $\mathcal{H} = \{\mathbb{1}\{x \geq t\} : t \in \mathbb{R}\}$ and we consider one target and two sources that all share the distribution $\mu_1$, while $\mu_0$ differs across them. All distributions are Gaussian with the same variance but different means. For source-1, $\alpha_{S_1} = \alpha$, which implies that it does not shrink $\hat{\mathcal{H}}_{\alpha,T}^*$ since $\hat{\mathcal{H}}_{S_1}(\hat{\alpha}_S) \supset \hat{\mathcal{H}}_{\alpha,T}^*$, and thus source-1 cannot reduce the target Type-I error. In contrast, for source-2, $\alpha_{S_2} > \alpha$, and $\hat{\mathcal{H}}' = \hat{\mathcal{H}}_S(\hat{\alpha}_{S_2}) \cap \hat{\mathcal{H}}_{\alpha,T}^*$ yields an improvement in the target Type-I error.*

## 5.2 COMPUTATIONAL GUARANTEE FOR THE TRANSFER LEARNING PROCEDURE

In this subsection, we provide a computational guarantee for the transfer learning procedure introduced in Section 4 by reformulating it as an optimization procedure. The optimization procedure consists of two stages: an auxiliary algorithm for computing $\hat{\alpha}_S$ in (6), and a main algorithm for solving the final model $\hat{h}$ in (8). For this optimization procedure, we consider a parameterized hypothesis class $\mathcal{H} \doteq \{h_\theta : \theta \in \mathbb{R}^d, \|\theta\| \leq B\}$.

**Notations.** $\|\cdot\|$ denotes the Euclidean ($\ell_2$) norm. In this subsection, $\hat{R}_{\varphi,\mu}(\theta)$ is a short hand for $\hat{R}_{\varphi,\mu}(h_\theta)$. The notation $f(\theta; z)$ denotes the evaluation of the function $f$ at $z$, parameterized by $\theta$.

We make the following assumption on the loss function $\varphi$ and the hypothesis class.

**Assumption 3** (Convexity, Bounded and Lipschitz Gradient)**.** *We assume that $\varphi((2y-1)h_\theta(x))$ is convex in $\theta$ for any $x \in \mathcal{X}$ and $y \in \{0,1\}$. In addition, the gradient is bounded, i.e., $\|\nabla_\theta \varphi((2y-1)h_\theta(x))\| \leq G$, for all $x \in \mathcal{X}$ and $\theta$ with $\|\theta\| \leq B$, and the gradient is Lipschitz continuous, i.e.,*

$$\|\nabla_\theta \varphi((2y-1)h_\theta(x)) - \nabla_\theta \varphi((2y-1)h_{\theta'}(x))\| \leq H\|\theta - \theta'\|,$$

*for all $x \in \mathcal{X}$, $y \in \{0,1\}$, and $\theta, \theta'$ with $\|\theta\|, \|\theta'\| \leq B$.*

The above assumption is standard and holds for common loss functions and hypothesis classes, such as the logistic loss, linear classifiers, and majority votes over a set of basis functions.

**First Stage: Computing $\hat{\alpha}_S$.** Define $g_{0,T}(\theta) \doteq \hat{R}_{\varphi,\mu_{0,T}}(\theta) - \alpha - \epsilon_{0,T}$, $g_{0,S}(\theta, \alpha') \doteq \hat{R}_{\varphi,\mu_{0,S}}(\theta) - \alpha' - \epsilon_{0,S}$, and $g_{1,T}(\theta) \doteq \hat{R}_{\varphi,\mu_{1,T}}(\theta) - \hat{R}_{\varphi,\mu_{1,T}}(\hat{\theta}_{T,\alpha-\epsilon_{0,T}}^*) - 6\epsilon_{1,T}$ where $\hat{\theta}_{T,\alpha-\epsilon_{0,T}}^* = \arg\min_\theta \hat{R}_{\varphi,\mu_{1,T}}(\theta)$ s.t. $g_{0,T}^-(\theta) \doteq \hat{R}_{\varphi,\mu_{0,T}}(\theta) - \alpha + \epsilon_{0,T} \leq 0$. By definition, $\hat{\alpha}_S$ is the smallest $\alpha' \geq \alpha$ such that the intersection of the sets $\{\theta : g_{0,T}(\theta) \leq 0\}$, $\{\theta : g_{0,S}(\theta, \alpha') \leq 0\}$ and $\{\theta : g_{1,T}(\theta) \leq 0\}$ is non-empty. This can be formulated as the following optimization problem:

$$\min_{\alpha',\theta} \alpha' \quad \text{s.t.} \quad \alpha' \geq \alpha, \; g'(\theta, \alpha') \leq 0. \tag{10}$$

where $g'(\theta, \alpha') \doteq \max\{g_{0,T}(\theta), g_{0,S}(\theta, \alpha'), g_{1,T}(\theta)\}$. Problem (10) is equivalent to the following minimax problem, in the sense that the primal solution of (11) coincides with the solution of (10) (Boyd & Vandenberghe, 2004, Section 5.2.3):

$$\min_{\alpha' \geq \alpha, \theta} \max_{\lambda \geq 0} \alpha' + \lambda g'(\theta, \alpha'). \tag{11}$$

---

**Algorithm 1:** CP-Solver$(f, g, \xi, \epsilon, \delta)$

**Input:** objective $f(\theta) \doteq \frac{1}{|\mathcal{Z}_f|} \sum_{\zeta \in \mathcal{Z}_f} f(\theta; \zeta)$, constraint $g(\theta) \doteq \frac{1}{|\mathcal{Z}_g|} \sum_{\zeta \in \mathcal{Z}_g} g(\theta; \zeta)$, slackness $\xi$, error tolerance $\epsilon$, failure probability $\delta$.

**Initialize:** $(\theta_0, \lambda_0) = (0, 0)$, $\eta = \frac{c_\eta}{\sqrt{N(\epsilon, \delta)}}$, $G = \sup_{\theta, \zeta_f, \zeta_g} \max\{\|\nabla f(\theta; \zeta_f)\|, \|\nabla g(\theta; \zeta_g)\|\}$,

$\gamma = G^2 \eta$, $\rho^2 \doteq B^2 + \frac{G^2}{r^2(g)}$, $C_g = \sup_{\theta, \zeta} |g(\theta; \zeta)|$ and

$$N(\epsilon, \delta) \gtrsim \left(G + C_g \sqrt{\log \frac{1}{\delta}}\right) \left(\frac{GC_g \sqrt{\log \frac{1}{\delta}} + \frac{G}{c_\eta}}{\epsilon^2 r^2(g)} + \frac{\frac{\rho^2}{c_\eta} + \frac{G}{r(g)} C_g \sqrt{\log \frac{1}{\delta}}}{\epsilon^2}\right).$$

**for** $t = 0, ..., N(\epsilon, \delta) - 1$ **do**
    Sample $\zeta_{f,t}, \zeta_{g,t}$ uniformly from $\mathcal{Z}_f$ and $\mathcal{Z}_g$, respectively
    $\theta_{t+1} = \theta_t - \eta(\nabla f(\theta_t; \zeta_{f,t})) + \lambda_t \nabla g(\theta_t; \zeta_{g,t}))$
    $\theta_{t+1} = \theta_{t+1} / \max\{1, \|\theta_{t+1}\| / B\}$
    $\lambda_{t+1} = [(1 - \gamma\eta)\lambda_t + \eta g(\theta_t; \zeta_{g,t})]_+$

**Output:** $\hat{\theta} = $ projection of $\frac{1}{N} \sum_{t=1}^{N} \theta_t$ onto the set $\{\theta : g(\theta) \leq -\xi\}$

---

To solve (11), we need an approximation of $\hat{\theta}^*_{T, \alpha - \epsilon_{0,T}}$. This can be obtained by solving: $\min_\theta \hat{R}_{\varphi, \mu_{1,T}}(\theta)$ s.t. $g^-_{0,T}(\theta) \leq 0$, which is equivalent to $\min_\theta \max_{\lambda \geq 0} \hat{R}_{\varphi, \mu_{1,T}}(\theta) + \lambda g^-_{0,T}(\theta)$. Following (Mahdavi et al., 2012), we solve this problem using a Stochastic Gradient Descent–Ascent (SGDA) method with a projection step, as described in Algorithm 1 CP-Solver. This algorithm solves convex programs which takes objective and constraint (possibly inexact) and returns an $\epsilon$ accurate solution. Since we only have access to inexact constraint $g$, the final projection step is onto $\{\theta : g(\theta) \leq -\xi\}$ to allow some slackness.

Once an approximation $\hat{\theta}_{T, \alpha - \epsilon_{0,T}}$ is obtained, we return to (11), replacing $g_{1,T}(\theta)$ with $\hat{g}_{1,T}(\theta) \doteq \hat{R}_{\varphi, \mu_{1,T}}(\theta) - \hat{R}_{\varphi, \mu_{1,T}}(\hat{\theta}_{T, \alpha - \epsilon_{0,T}}) - 6\epsilon_{1,T}$, and again call Algorithm 1 to compute an approximation of $\hat{\alpha}_S$. The following key quantity will appear in the analysis of Algorithm 1.

**Definition 5.** *We define* $r(g(\theta)) \doteq \inf\{\|v\| : v \in \partial g(\theta), g(\theta) = 0, \theta \in \text{dom}(g)\}$.

The above notion, introduced by Mahdavi et al. (2012), measures the magnitude of the constraint gradient along the boundary of the constraint set. It is used in the convergence analysis to control the distance between the returned solution and the solution prior to the final projection step.

**Second Stage: Solving the Final Model** $\hat{h}$. We reduce the final step of the procedure in Section 4—which checks whether $\hat{\mathcal{H}}'_{1,S}$ and $\hat{\mathcal{H}}'_{1,T}$ intersect, as defined in (8)—to a constrained optimization problem. We define $g_{\hat{\alpha}_S}(\theta) \doteq \max\{g_{0,T}(\theta), g_{0,S}(\theta, \hat{\alpha}_S), g_{1,T}(\theta)\}$. The set $\{\theta : g_{\hat{\alpha}_S}(\theta) \leq 0\}$ coincides with the hypothesis class $\hat{\mathcal{H}}'$ defined in Section 4. We also define

$$g'_D(\theta) \doteq \hat{R}_{\varphi, \mu_{1,D}}(\theta) - \hat{R}_{\varphi, \mu_{1,D}}(\hat{\theta}^*_{D, \hat{\alpha}_S}) - 2\epsilon_{1,D} \tag{12}$$

where $\hat{\theta}^*_{D, \hat{\alpha}_S} = \arg\min_\theta \hat{R}_{\varphi, \mu_{1,D}}(\theta)$ s.t. $g_{\hat{\alpha}_S}(\theta) \leq 0$ for $D \in \{S, T\}$. The set $\{\theta : g_{\hat{\alpha}_S}(\theta) \leq 0, g'_D(\theta) \leq 0\}$ coincides with the hypothesis class $\hat{\mathcal{H}}'_{1,D}$ as in (8). Then, the second stage of the algorithm is to solve the following problem:

$$\min_\theta \hat{R}_{\varphi, \mu_{1,S}}(\theta) \quad \text{s.t.} \quad g_{S,T}(\theta) \leq 0 \tag{13}$$

---

**Algorithm 2:** NP-Transfer-Learning

---

**Initialize:** $G_\lambda \leftarrow 2C + \max\{\epsilon_{0,S}, \epsilon_{0,T}, \epsilon_{1,S}, \epsilon_{1,T}\}$

$r_T \leftarrow r\big(g_{0,T}^-(\theta)\big), r_{\hat{\alpha}_S} \leftarrow r(g'(\theta, \alpha')), r_S' \leftarrow r(g_S'(\theta)), r_T' \leftarrow r(g_T'(\theta)), r_{S,T} \leftarrow r(g_{S,T}(\theta))$

**Define the slackness function:** $\xi(\epsilon, r) \doteq \min\left\{ \frac{1}{4H}, \left(G + G_\lambda \sqrt{\log\frac{1}{\delta}}\right)^{-1} \cdot \frac{\epsilon}{2 + 2H\epsilon} \right\} r.$

**Tolerance setup:**

$\epsilon_{S,T} \leftarrow \epsilon_{1,S}, \quad \epsilon_T' \leftarrow \min\{\xi(\epsilon_{S,T}, r_{S,T}), \epsilon_{1,T}\}, \quad \epsilon_S' \leftarrow \epsilon_{1,S}$

$\epsilon_{\hat{\alpha}_S} \leftarrow \min\{\xi(\epsilon_{S,T}, r_{S,T}), \xi(\epsilon_S', r_S'), \xi(\epsilon_T', r_T')\}$

$\epsilon_{T,\alpha-\epsilon_{0,T}} \leftarrow \min\{\xi(\epsilon_{S,T}, r_{S,T}), \xi(\epsilon_S', r_S'), \xi(\epsilon_T', r_T'), \xi(\epsilon_{\hat{\alpha}_S}, r_{\hat{\alpha}_S})\}$

**Warm-start on $T$:**

$\hat{\theta}_{T,\alpha-\epsilon_{0,T}} \leftarrow \texttt{CP-Solver}\left(\hat{R}_{\varphi,\mu_{1,T}}, g_{0,T}^-, 0, \epsilon_{T,\alpha-\epsilon_{0,T}}, \frac{\delta}{5}\right)$

$\hat{g}_{1,T}(\theta) \leftarrow \hat{R}_{\varphi,\mu_{1,T}}(\theta) - \hat{R}_{\varphi,\mu_{1,T}}(\hat{\theta}_{T,\alpha-\epsilon_{0,T}}) - 6\epsilon_{1,T}, \hat{g}'(\theta, \alpha) \leftarrow \max\{g_{0,T}(\theta), g_{0,S}(\theta, \alpha), \hat{g}_{1,T}(\theta)\}$

**Compute $\hat{\alpha}$:** $(\hat{\alpha}, \perp) \leftarrow \texttt{CP-Solver}\left(\alpha', \hat{g}'(\theta, \alpha'), \xi(\epsilon_{\hat{\alpha}_S}, r_{\hat{\alpha}_S}), \epsilon_{\hat{\alpha}_S}, \frac{\delta}{5}\right)$

**Define the joint constraint:** $\hat{g}_{\hat{\alpha}}(\theta) \leftarrow \max\{g_{0,T}(\theta), g_{0,S}(\theta, \hat{\alpha}), \hat{g}_{1,T}(\theta)\}$

**Solve $T$ and $S$ subproblems:**

$\hat{\theta}_T' \leftarrow \texttt{CP-Solver}\left(\hat{R}_{\varphi,\mu_{1,T}}, \hat{g}_{\hat{\alpha}}, \xi(\epsilon_T', r_T'), \epsilon_T', \frac{\delta}{5}\right)$

$\hat{\theta}_S' \leftarrow \texttt{CP-Solver}\left(\hat{R}_{\varphi,\mu_{1,S}}, \hat{g}_{\hat{\alpha}}, \xi(\epsilon_S', r_S'), \epsilon_S', \frac{\delta}{5}\right)$

**Define the final constraint:**

$\hat{g}_T'(\theta) \leftarrow \hat{R}_{\varphi,\mu_{1,T}}(\theta) - \hat{R}_{\varphi,\mu_{1,T}}(\hat{\theta}_T') - 2\epsilon_{1,T} ; \hat{g}_{S,T}(\theta) \leftarrow \max\{\hat{g}_{\hat{\alpha}}(\theta), \hat{g}_T'(\theta)\}$

**Final solve $\hat{h}$:**

$\tilde{\theta} \leftarrow \texttt{CP-Solver}\left(\hat{R}_{\varphi,\mu_{1,S}}, \hat{g}_{S,T}, \xi(\epsilon_{S,T}, r_{S,T}), \epsilon_{S,T}, \frac{\delta}{5}\right)$

**Output:** If $\hat{R}_{\varphi,\mu_{1,S}}(\tilde{\theta}) - \hat{R}_{\varphi,\mu_{1,S}}(\hat{\theta}_S') > 2\epsilon_{1,S}$: $\hat{\theta} \leftarrow \hat{\theta}_T'$ **else:** $\hat{\theta} \leftarrow \tilde{\theta}$

---

where $g_{S,T}(\theta) \doteq \max\{g_{\hat{\alpha}_S}(\theta), g_T'(\theta)\}$. To approximate $\hat{\theta}_{D,\hat{\alpha}_S}^*$ in (12) for $D \in \{S, T\}$, we again call Algorithm 1 to solve $\min_{\theta: g_{\hat{\alpha}_S}(\theta) \leq 0} \hat{R}_{\varphi,\mu_{1,D}}(\theta)$ to a certain accuracy prescribed in Algorithm 2. At last, if the approximate solution of (13) also has small source empirical Type-II error, we return it; otherwise, we simply return target model $\hat{\theta}_T'$. The whole algorithm (first stage + second stage) is presented in Algorithm 2.

**Theorem 2.** *Let Assumption 3 hold. With properly chosen $c_\eta$ in Algorithm 1, then the Algorithm 2 outputs $\hat{\theta}$ such that, with probability at least $1 - 5\delta$, the hypothesis $h_{\hat{\theta}}$ satisfies the statistical guarantee in Theorem 1, and the number of stochastic gradient evaluations is bounded by*

$$c \cdot \left(G + G_\lambda \sqrt{\log\frac{1}{\delta}}\right)^3 \left(\frac{GG_\lambda \sqrt{\log\frac{1}{\delta}} + \frac{G}{c_\eta}}{r_{\min}^2} + \frac{B^2 + \frac{G^2}{r_{\min}^2}}{c_\eta} + \frac{G}{r_{\min}} G_\lambda \sqrt{\log\frac{1}{\delta}}\right)^2 \cdot \frac{1}{\epsilon_{\min}^2}$$

*for some $c > 0$, where $\epsilon_{\min} \doteq \min\{\epsilon_{1,S}, \epsilon_{1,T}, \xi(\epsilon_{S,T}, r_{S,T}), \xi(\epsilon_S', r_S'), \xi(\epsilon_T', r_T'), \xi(\epsilon_{\hat{\alpha}_S}, r_{\hat{\alpha}_S})\}$, and $r_{\min} \doteq \min\{r_T, r_{\hat{\alpha}_S}, r_T', r_S', r_{S,T}\}$ where $\xi$ is defined in Algorithm 2.*

**Remark 5** (Computational Complexity)**.** *The proposed algorithm outputs a model that satisfies the learning guarantee as in Theorem 1, and the total number of (stochastic) gradient evaluations scales as $\min\{\frac{1}{\epsilon_{1,S}}, \frac{1}{\epsilon_{1,T}}\}$. Roughly speaking, we call* $\texttt{CP-Solver}$ *procedures multiple times to solve different convex programs to different accuracies, and the final gradient complexity is dominated by the worst one of them.*

## 6 EXPERIMENTS

In this section, we implement the proposed algorithm on two climate datasets: Yu et al. (2023), with 124 features, and NASA POWER (2024), with 6 features, for heavy rain detection, where source and target correspond to different locations. We use a multilayer perceptron with two hidden layers and

ReLU activation. In these experiments, the number of target samples is fixed at $n_{0,T} = n_{1,T} = 40$, while the number of source samples $n_{0,S} = n_{1,S}$ varies from 50 to 950, with $\alpha = 0.1$. Each case is run for 10 trials, and the trained model is evaluated on a target test set of size $n_{0,T} = n_{1,T} = 1700$.

In Figure 2, the proposed Transfer Learning Algorithm (TLA) keeps the Type-I error close to the threshold, whereas the Only Target method exceeds it more and also performs poorly on the Type-II error. In this figure, the source is informative, and TLA leverages it to outperform Only Target method without requiring prior knowledge of the relatedness between source and target.

In Figure 3, the source is not informative, and the Only Source method suffers from a high Type-II error. TLA, however, adaptively avoids negative transfer and achieves performance comparable to the Only Target method, corroborating the result in Theorem 1. Hence, when the source is informative, our algorithm adaptively exploits it, and when it is not, it avoids negative transfer, even in the presence of potential shifts in both class-0 and class-1 distributions.

Moreover, we conduct additional experiments on synthetic Gaussian data, provided in Appendix D.

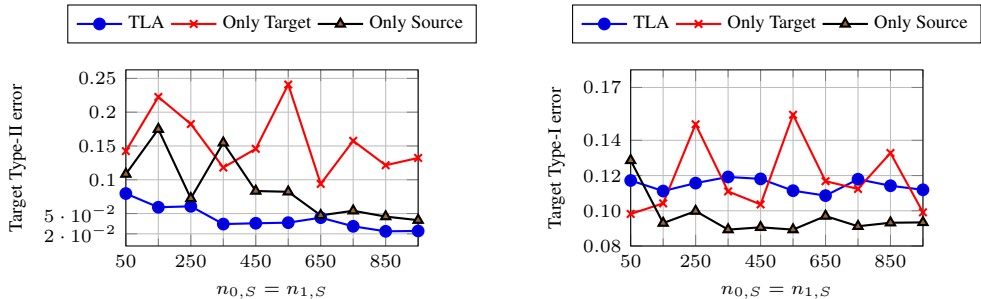

Figure 2: The performance of our algorithm (TLA), along with two baselines—using only source data and only target data—on Climate data (Yu et al., 2023), is evaluated under a Type-I error threshold of $\alpha = 0.1$. In this experiment, we fix $n_{0,T} = n_{1,T} = 40$ and vary the number of source samples $n_{0,S} = n_{1,S}$.

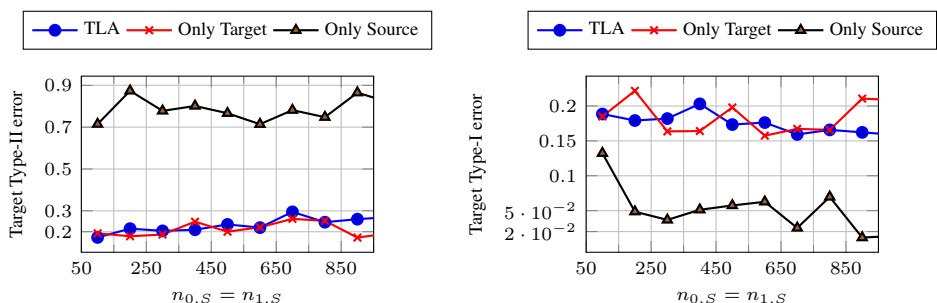

Figure 3: The performance of our algorithm (TLA), along with two baselines—using only source data and only target data—on Climate data (NASA POWER, 2024), is evaluated under a Type-I error threshold of $\alpha = 0.1$. In this experiment, we fix $n_{0,T} = n_{1,T} = 40$ and vary the number of source samples $n_{0,S} = n_{1,S}$.

## 7    CONCLUSION AND FUTURE DIRECTIONS

This work introduces an adaptive transfer learning procedure for NP classification under distribution shifts in both class-0 and class-1 distributions. The method provides statistical guarantees that recover existing results when only the class-1 distribution shifts, and it additionally offers computational guarantees through a reduction to convex programs. The procedure adapts to the unknown relatedness between source and target, improving performance when the source is informative while matching the target-only method when it is not. A natural future direction is to establish matching lower bounds for the general setting in which both class-0 and class-1 distributions may shift. Such lower bounds complement the upper bounds in this work and clarify the fundamental limits of transfer in NP classification.

ACKNOWLEDGMENT

Samory Kpotufe and Yuyang Deng were funded by the NSF through the Learning the Earth with Artificial Intelligence and Physics (LEAP) Science and Technology Center (STC) (Award #2019625).

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

## A    EXPRESSING THEOREM 1 IN TERMS OF TRANSFER EXPONENT

Theorem 1 uses the functions $\phi_1^{S \to T}$ and $\phi_0^{S \to T}$ to capture the source effect in the generaliztion bound. This formulation is broad enough to yield bounds expressed through the notion of transfer exponent (Hanneke & Kpotufe, 2019; Kalan & Kpotufe, 2024b).

Regarding the $\varphi$-Type-II excess error, assume that there exist constants $C_1 > 0$ and $\rho_1 > 0$ such that for all $h \in \mathcal{H}$,

$$\left[ \mathcal{E}_{1,S}(h \mid h^*_{S,T,\alpha}) \right]_+ \ \geq \ C_1 \left[ \mathcal{E}_{1,T}(h \mid h^*_{S,T,\alpha}) \right]_+^{\rho_1}.$$

Then, by Definition 4, for any $\varepsilon \geq 0$ we have $\phi_1^{S \to T}(\varepsilon) \ \leq \ C_1^{1/\rho_1} \varepsilon^{1/\rho_1}$, and consequently the term $\phi_1^{S \to T}(c \cdot \epsilon_{1,S})$ in Theorem 1 can be replaced with $c \cdot \epsilon_{1,S}^{1/\rho_1}$ for some numerical constant $c$. Regarding the $\varphi$-Type-I error, first we define the minimal source level that guarantees $\alpha$ level on target as

$$\alpha^*_{S \to T} \ := \ \inf\{\alpha' \in [0,1] : \ \phi_0^{S \to T}(\alpha') \leq \alpha\}.$$

Assume that there exist constants $C_0 > 0$ and $\rho_0 > 0$ such that for all $h \in \mathcal{H}$

$$\left[ R_{\varphi,\mu_{0,S}}(h) - \alpha^*_{S \to T}(\alpha) \right]_+ \ \geq \ C_0 \left[ R_{\varphi,\mu_{0,T}}(h) - \alpha \right]_+^{\rho_0}.$$

Then the $\varphi$-Type-I error bound takes the form

$$R_{\varphi,\mu_{0,T}}(\hat{h}) \ \leq \ \begin{cases} \alpha + \min\left\{ \epsilon_{0,T}, \ c \cdot \left( \left[ \alpha + \epsilon_{0,S} - \alpha^*_{S \to T}(\alpha) \right]_+ \right)^{1/\rho_0} \right\}, & \alpha \geq \alpha_S, \\[2ex] \alpha + \min\left\{ \epsilon_{0,T}, \ c \cdot \left( \left[ \hat{\alpha}_S + \epsilon_{0,S} - \alpha^*_{S \to T}(\alpha) \right]_+ \right)^{1/\rho_0} \right\}, & \alpha < \alpha_S. \end{cases}$$

In particular, if $\mu_{0,S} = \mu_{0,T}$ then $\alpha^*_{S \to T} = \alpha$, and the bound matches that of Kalan et al. (2025); Kalan & Kpotufe (2024b).

## B    PROOF OF THEOREM 1

We begin with two lemmas. The first establishes a uniform concentration bound for the difference between empirical and population errors. The second shows that the $\varphi$-Type-II error can be controlled by the deviation in the $\varphi$-Type-I error, provided that both the hypothesis class $\mathcal{H}$ and the loss $\varphi$ are convex.

**Lemma 1.** *Let $\delta > 0$ and let $\mathcal{H}$ be a hypothesis class satisfying Assumption 1. Denote by $\hat{R}_{\varphi,\mu}$ the empirical error computed from $n$ i.i.d. samples drawn from a distribution $\mu$, where $\mu$ may be either $\mu_0$ or $\mu_1$. Then, with probability at least $1 - \delta$,*

$$\sup_{h \in \mathcal{H}} \left| R_{\varphi,\mu}(h) - \hat{R}_{\varphi,\mu}(h) \right| \ \leq \ \frac{4 B_{\mathcal{H}} L + C \sqrt{2 \log(2/\delta)}}{\sqrt{n}},$$

*where the constant $C$ is as defined in Definition 1.*

*Proof.* See the proof of Proposition 1 in Kalan et al. (2025), which relies on symmetrization and McDiarmid's inequality (Shalev-Shwartz & Ben-David, 2014, Chapter 26). $\qquad\square$

**Lemma 2.** *Suppose that the hypothesis class $\mathcal{H}$ satisfies Assumption 2, and that the surrogate loss $\varphi$ in Definition 1 is convex. Further assume that $\mathcal{H}_T(\alpha/8)$ is nonempty and that $\epsilon_{0,T} \leq \frac{7\alpha}{8}$. Then,*

$$R_{\varphi,\mu_{1,T}}\left( h^*_{T,\alpha-\epsilon_{0,T}} \right) - R_{\varphi,\mu_{1,T}}\left( h^*_{T,\alpha} \right) \ \leq \ c \cdot \epsilon_{0,T}, \tag{14}$$

*for some numerical constant $c$.*

*Proof.* See the proof of Theorem 2 in Kalan et al. (2025), which shows that

$$\gamma(\alpha) \ := \ \inf_{h \in \mathcal{H}_T(\alpha)} R_{\varphi,\mu_{1,T}}(h)$$

is a non-increasing convex function on $[0,1]$. $\qquad\square$

Consider the event that Lemma 1 holds simultaneously for the distributions $\mu_{i,D}$ with $i \in \{0,1\}$ and $D \in \{S,T\}$; this event occurs with probability at least $1 - 4\delta$. We divide the proof into two cases: (i) $\alpha \geq \alpha_S$, and (ii) $\alpha < \alpha_S$.

**Case I:** $\alpha \geq \alpha_S$.

First, we show that $\hat{\alpha}_S = \alpha$. From (6), we know that $\hat{\alpha}_S \geq \alpha$. Hence, it suffices to establish that $\hat{\mathcal{H}}_S(\alpha) \cap \hat{\mathcal{H}}_{\alpha,T}^* \neq \emptyset$. Under the considered event, we have

$$\hat{\mathcal{H}}_S(\alpha) \supseteq \mathcal{H}_S(\alpha) \supseteq \mathcal{H}_S(\alpha_S),$$

so by (4) there exists $h_{T,\alpha}^* \in T^*(\alpha)$ such that $h_{T,\alpha}^* \in \hat{\mathcal{H}}_S(\alpha)$. On the other hand, $h_{T,\alpha}^* \in \hat{\mathcal{H}}_{\alpha,T}^*$, since

$$\hat{R}_{\varphi,\mu_{0,T}}(h_{T,\alpha}^*) \leq R_{\varphi,\mu_{0,T}}(h_{T,\alpha}^*) + \epsilon_{0,T} \leq \alpha + \epsilon_{0,T} \tag{15}$$

and

$$\hat{R}_{\varphi,\mu_{1,T}}(h_{T,\alpha}^*) \leq R_{\varphi,\mu_{1,T}}(h_{T,\alpha}^*) + \epsilon_{1,T} \leq R_{\varphi,\mu_{1,T}}(h_{T,\alpha-\epsilon_{0,T}}^*) + \epsilon_{1,T}$$
$$\leq \hat{R}_{\varphi,\mu_{1,T}}(h_{T,\alpha-\epsilon_{0,T}}^*) + 2\epsilon_{1,T}, \tag{16}$$

which implies that

$$h_{T,\alpha}^* \in \hat{\mathcal{H}}_S(\alpha) \cap \hat{\mathcal{H}}_{\alpha,T}^* \quad \text{and} \quad \hat{\mathcal{H}}_S(\alpha) \cap \hat{\mathcal{H}}_{\alpha,T}^* \neq \emptyset. \tag{17}$$

Regarding the $\varphi$-Type-I error bound in Theorem 1, we have $\hat{h} \in \hat{\mathcal{H}}_{1,T}' \subseteq \hat{\mathcal{H}}' = \hat{\mathcal{H}}_S(\alpha) \cap \hat{\mathcal{H}}_{\alpha,T}^*$, which implies that

$$R_{\varphi,\mu_{0,S}}(\hat{h}) \leq \alpha + 2\epsilon_{0,S} \quad \text{and} \quad R_{\varphi,\mu_{0,T}}(\hat{h}) \leq \alpha + 2\epsilon_{0,T}.$$

Together with Definition 4, this yields the bound stated in Theorem 1.

Regarding the $\varphi$-Type-II excess error bound in Theorem 1, first consider the case $h_{S,T,\alpha}^* \notin \hat{\mathcal{H}}'$. Since $h_{S,T,\alpha}^* \in \mathcal{H}_S(\alpha) \subseteq \hat{\mathcal{H}}_S(\alpha)$ and $h_{S,T,\alpha}^* \in \mathcal{H}_T(\alpha) \subseteq \hat{\mathcal{H}}_T(\alpha)$, it follows from (5) that

$$\hat{R}_{\varphi,\mu_{1,T}}(h_{S,T,\alpha}^*) > \hat{R}_{\varphi,\mu_{1,T}}(\hat{h}_{T,\alpha-\epsilon_{0,T}}^*) + 6\epsilon_{1,T}. \tag{18}$$

Thus,

$$\epsilon_{1,T} + R_{\varphi,\mu_{1,T}}(h_{S,T,\alpha}^*) \geq \hat{R}_{\varphi,\mu_{1,T}}(h_{S,T,\alpha}^*) > \hat{R}_{\varphi,\mu_{1,T}}(\hat{h}_{T,\alpha-\epsilon_{0,T}}^*) + 6\epsilon_{1,T}$$
$$\geq R_{\varphi,\mu_{1,T}}(\hat{h}_{T,\alpha-\epsilon_{0,T}}^*) + 5\epsilon_{1,T}$$
$$\geq R_{\varphi,\mu_{1,T}}(h_{T,\alpha}^*) + 5\epsilon_{1,T} \tag{19}$$

where the last inequality uses the fact that

$$R_{\varphi,\mu_{0,T}}(\hat{h}_{T,\alpha-\epsilon_{0,T}}^*) \leq \hat{R}_{\varphi,\mu_{0,T}}(\hat{h}_{T,\alpha-\epsilon_{0,T}}^*) + \epsilon_{0,T} \leq \alpha$$

which implies $\hat{h}_{T,\alpha-\epsilon_{0,T}}^* \in \mathcal{H}_T(\alpha)$, together with the definition $h_{T,\alpha}^* = \underset{h \in \mathcal{H}_T(\alpha)}{\arg\min} R_{\varphi,\mu_{1,T}}(h)$. Therefore,

$$R_{\varphi,\mu_{1,T}}(h_{S,T,\alpha}^*) - R_{\varphi,\mu_{1,T}}(h_{T,\alpha}^*) > 4\epsilon_{1,T} \tag{20}$$

On the other hand, for every $h \in \hat{\mathcal{H}}_{1,T}'$ we have

$$R_{\varphi,\mu_{1,T}}(h) \leq \hat{R}_{\varphi,\mu_{1,T}}(h) + \epsilon_{1,T} \leq \hat{R}_{\varphi,\mu_{1,T}}^*(\hat{\mathcal{H}}') + 3\epsilon_{1,T} \overset{(1)}{\leq} \hat{R}_{\varphi,\mu_{1,T}}(h_{T,\alpha}^*) + 3\epsilon_{1,T}$$
$$\leq R_{\varphi,\mu_{1,T}}(h_{T,\alpha}^*) + 4\epsilon_{1,T},$$

where inequality (1) holds because $h_{T,\alpha}^* \in \hat{\mathcal{H}}'$ by (17), and $\hat{R}_{\varphi,\mu_{1,T}}^*(\hat{\mathcal{H}}') = \underset{h \in \hat{\mathcal{H}}'}{\min} \hat{R}_{\varphi,\mu_{1,T}}(h)$. Hence,

$$\forall h \in \hat{\mathcal{H}}_{1,T}', \quad \mathcal{E}_{1,T}(h) \leq 4\epsilon_{1,T}. \tag{21}$$

Combining this with (20), we conclude the $\varphi$-Type-II excess error bound in Theorem 1 for the case where $h^*_{S,T,\alpha} \notin \hat{\mathcal{H}}'$.

Next, consider the case $h^*_{S,T,\alpha} \in \hat{\mathcal{H}}'$. We first analyze the case where $\hat{\mathcal{H}}'_{1,S} \cap \hat{\mathcal{H}}'_{1,T} \neq \emptyset$, in which the learner selects $\hat{h} \in \hat{\mathcal{H}}'_{1,S} \cap \hat{\mathcal{H}}'_{1,T}$. By (21), we know that $\mathcal{E}_{1,T}(\hat{h}) \leq 4\epsilon_{1,T}$. Moreover, since $\hat{h} \in \hat{\mathcal{H}}'_{1,S}$, we have

$$R_{\varphi,\mu_{1,S}}(\hat{h}) \leq \hat{R}_{\varphi,\mu_{1,S}}(\hat{h}) + \epsilon_{1,S} \leq \hat{R}^*_{\varphi,\mu_{1,S}}(\hat{\mathcal{H}}') + 3\epsilon_{1,S}$$

$$\overset{(1)}{\leq} \hat{R}_{\varphi,\mu_{1,S}}(h^*_{S,T,\alpha}) + 3\epsilon_{1,S}$$

$$\leq R_{\varphi,\mu_{1,S}}(h^*_{S,T,\alpha}) + 4\epsilon_{1,S} \tag{22}$$

where inequality (1) holds because $h^*_{S,T,\alpha} \in \hat{\mathcal{H}}'$, and $\hat{R}^*_{\varphi,\mu_{1,S}}(\hat{\mathcal{H}}') = \min_{h \in \hat{\mathcal{H}}'} \hat{R}_{\varphi,\mu_{1,S}}(h)$. Hence,

$$\mathcal{E}_{1,S}(h|h^*_{S,T,\alpha}) = R_{\varphi,\mu_{1,S}}(\hat{h}) - R_{\varphi,\mu_{1,S}}(h^*_{S,T,\alpha}) \leq 4\epsilon_{1,S}, \tag{23}$$

which, together with Definition 4, implies

$$\mathcal{E}_{1,T}(\hat{h}) \leq R_{\varphi,\mu_{1,T}}(h^*_{S,T,\alpha}) - R_{\varphi,\mu_{1,T}}(h^*_{T,\alpha}) + \phi_1^{S \to T}(4\epsilon_{1,S}). \tag{24}$$

This establishes the $\varphi$-Type-II excess error bound in Theorem 1. In the case $\hat{\mathcal{H}}'_{1,S} \cap \hat{\mathcal{H}}'_{1,T} = \emptyset$, the learner selects $\hat{h} = \arg\min_{h \in \hat{\mathcal{H}}'} \hat{R}_{\varphi,\mu_{1,T}}(h)$. Let $\hat{h}'_S$ denote a function belonging to the set $\hat{\mathcal{H}}'_{1,S}$. Since $\hat{\mathcal{H}}'_{1,S} \cap \hat{\mathcal{H}}'_{1,T} = \emptyset$, we obtain

$$\hat{R}_{\varphi,\mu_{1,T}}(\hat{h}'_S) > \hat{R}_{\varphi,\mu_{1,T}}(\hat{h}) + 2\epsilon_{1,T}. \tag{25}$$

We claim that $R_{\varphi,\mu_{1,T}}(\hat{h}'_S) > R_{\varphi,\mu_{1,T}}(\hat{h})$. Otherwise, we would have

$$\hat{R}_{\varphi,\mu_{1,T}}(\hat{h}'_S) \leq R_{\varphi,\mu_{1,T}}(\hat{h}'_S) + \epsilon_{1,S} \leq R_{\varphi,\mu_{1,T}}(\hat{h}) + \epsilon_{1,S} \leq \hat{R}_{\varphi,\mu_{1,T}}(\hat{h}) + 2\epsilon_{1,S}$$

which contradicts (25). Therefore,

$$\mathcal{E}_{1,T}(\hat{h}) = R_{\varphi,\mu_{1,T}}(\hat{h}) - R_{\varphi,\mu_{1,T}}(h^*_{T,\alpha})$$

$$< R_{\varphi,\mu_{1,T}}(\hat{h}'_S) - R_{\varphi,\mu_{1,T}}(h^*_{T,\alpha})$$

$$\leq R_{\varphi,\mu_{1,T}}(h^*_{S,T,\alpha}) - R_{\varphi,\mu_{1,T}}(h^*_{T,\alpha}) + \phi_1^{S \to T}(4\epsilon_{1,S}). \tag{26}$$

where the last inequality follows from (24), which holds for every $h \in \hat{\mathcal{H}}'_{1,S}$. Hence, together with $\mathcal{E}_{1,T}(\hat{h}) \leq 4\epsilon_{1,T}$, we conclude the $\varphi$-Type-II excess error bound in Theorem 1.

**Case II:** $\alpha < \alpha_S$.

First, we show that $\hat{\alpha}_S \leq \alpha_S$. Since $\alpha_S > \alpha$, by (6) it suffices to show that $\hat{\mathcal{H}}_S(\alpha_S) \cap \hat{\mathcal{H}}^*_{\alpha,T} \neq \emptyset$. By (4), there exists $h^*_{T,\alpha} \in T^*(\alpha)$ such that $h^*_{T,\alpha} \in \mathcal{H}_S(\alpha_S) \subseteq \hat{\mathcal{H}}_S(\alpha_S)$. Moreover, using (15) and (16), we have $h^*_{T,\alpha} \in \hat{\mathcal{H}}^*_{\alpha,T}$, which completes the argument.

Regarding the $\varphi$-Type-I error bound in Theorem 1, we have $\hat{h} \in \hat{\mathcal{H}}'_{1,T} \subseteq \hat{\mathcal{H}}' = \hat{\mathcal{H}}_S(\hat{\alpha}_S) \cap \hat{\mathcal{H}}^*_{\alpha,T}$, which implies that

$$R_{\varphi,\mu_{0,S}}(\hat{h}) \leq \hat{\alpha}_S + 2\epsilon_{0,S} \quad \text{and} \quad R_{\varphi,\mu_{0,T}}(\hat{h}) \leq \alpha + 2\epsilon_{0,T}.$$

Together with Definition 4, this yields the bound stated in Theorem 1.

Regarding the $\varphi$-Type-II excess error bound in Theorem 1, first consider the case $h^*_{S,T,\alpha} \notin \hat{\mathcal{H}}'$. Since $h^*_{S,T,\alpha} \in \mathcal{H}_S(\alpha) \subseteq \mathcal{H}_S(\hat{\alpha}_S) \subseteq \hat{\mathcal{H}}_S(\hat{\alpha}_S)$ and $h^*_{S,T,\alpha} \in \mathcal{H}_T(\alpha) \subseteq \hat{\mathcal{H}}_T(\alpha)$, it follows, by the same reasoning as in (18) and (19), that (20) holds. On the other hand, for all $h \in \hat{\mathcal{H}}^*_{\alpha,T}$ we have

$$R_{\varphi,\mu_{1,T}}(h) \leq \hat{R}_{\varphi,\mu_{1,T}}(h) + \epsilon_{1,T} \leq \hat{R}_{\varphi,\mu_{1,T}}(h^*_{T,\alpha-\epsilon_{0,T}}) + 7\epsilon_{1,T}$$

$$\leq \hat{R}_{\varphi,\mu_{1,T}}(h^*_{T,\alpha-2\epsilon_{0,T}}) + 7\epsilon_{1,T}$$

$$\leq R_{\varphi,\mu_{1,T}}(h^*_{T,\alpha-2\epsilon_{0,T}}) + 8\epsilon_{1,T}$$

$$\leq R_{\varphi,\mu_{1,T}}(h^*_{T,\alpha}) + c \cdot \epsilon_{1,T}$$

for some numerical constant $c$, where in the last inequality we used Lemma 2 together with the assumptions $\epsilon_{0,T} \leq \frac{7\alpha}{16}$ and $n_{0,T} \geq n_{1,T}$. Therefore, for all $h \in \hat{\mathcal{H}}'_{1,T} \subseteq \hat{\mathcal{H}}' \subseteq \hat{\mathcal{H}}^*_{\alpha,T}$ we obtain

$$\mathcal{E}_{1,T}(h) \leq c \cdot \epsilon_{1,T} \tag{27}$$

which, together with (20), yields the bound stated in Theorem 1 for the case where $h^*_{S,T,\alpha} \notin \hat{\mathcal{H}}'$.

Next, consider the case $h^*_{S,T,\alpha} \in \hat{\mathcal{H}}'$. We first analyze the case where $\hat{\mathcal{H}}'_{1,S} \cap \hat{\mathcal{H}}'_{1,T} \neq \emptyset$, in which the learner selects $\hat{h} \in \hat{\mathcal{H}}'_{1,S} \cap \hat{\mathcal{H}}'_{1,T}$. By (27), we have $\mathcal{E}_{1,T}(h) \leq c \cdot \epsilon_{1,T}$. Furthermore, since $\hat{h} \in \hat{\mathcal{H}}'_{1,S}$ and $h^*_{S,T,\alpha} \in \hat{\mathcal{H}}'$, we can apply the same reasoning as in (22) and (23) to obtain (24), which in turn establishes the $\varphi$-Type-II excess error bound in Theorem 1.

In the case $\hat{\mathcal{H}}'_{1,S} \cap \hat{\mathcal{H}}'_{1,T} = \emptyset$, the learner selects $\hat{h} = \arg\min_{h \in \hat{\mathcal{H}}'} \hat{R}_{\varphi,\mu_{1,T}}(h)$. The same reasoning as in the case $\alpha \geq \alpha_S$ yields (26), and together with (27), this implies the $\varphi$-Type-II excess error bound in Theorem 1.

## C   PROOF OF THEOREM 2

In this section we will present the proof of Theorem 2, the convergence rate of our optimization procedure. We first introduce the following Theorem that establishes the convergence of our convex program solver (Algorithm 1), which is an important component of our main algorithm.

**Theorem 3.** *Let $\theta^* = \arg\min_\theta f(\theta)$ s.t. $\tilde{g}(\theta) \leq 0$ and $\rho^2 \doteq B^2 + \frac{G^2}{r^2}$. Assume $f$ and $g$ are convex, $G$ Lipschitz, $H$ smooth and $\sup_{\theta,\zeta} |g(\theta;\zeta)| \leq C_g$. Suppose $\epsilon_0 \leq \frac{C_g\sqrt{3\log\frac{2}{\delta}}}{(\lambda^* + \sqrt{2}\rho)\sqrt{N}}$. For Algorithm 1, assuming $g(\theta) = \tilde{g}(\theta) - c$ for some $c \in [0, \epsilon_0]$, if we choose $\eta = \frac{c_\eta}{\sqrt{N}}$, where*

$$c_\eta \leq \min\left\{ \frac{\rho}{2\sqrt{3}C_g}, \frac{\rho}{12\sqrt{6}(1 + \sqrt{2}\rho + \lambda^*)G\sqrt{3\log\frac{2}{\delta}}}, \frac{\rho}{12\sqrt{6}C_g\sqrt{\log\frac{2}{\delta}}}, \frac{\rho}{32\sqrt{N}\epsilon_0}, \frac{\sqrt{G}}{r - 2H\epsilon_0} \right\},$$

*then with probability at least $1 - \delta$ we have*

$$f(\hat{\theta}) - f(\theta^*) \lesssim \left( G + C_g\sqrt{\log\frac{1}{\delta}} \right) \left( \frac{GC_g\sqrt{\log\frac{1}{\delta}} + \frac{G}{c_\eta}}{\sqrt{N}(r - 2H\epsilon_0)^2} + \frac{\xi + \epsilon_0}{(r - 2H\epsilon_0)} + \frac{\frac{\rho^2}{c_\eta} + \frac{G}{r}C_g\sqrt{\log\frac{1}{\delta}}}{\sqrt{N}} \right).$$

*That is, if we choose $\xi = \epsilon_0$ and $\epsilon_0 = \min\left\{ \frac{r\epsilon}{1+H\epsilon}\left( G + C_g\sqrt{\log\frac{1}{\delta}} \right)^{-1}, \frac{r}{4H} \right\}$, we know $f(\hat{\theta}) - f(\theta^*) \leq \epsilon$ and the number of total gradient evaluations is bounded by*

$$N \gtrsim \left( G + C_g\sqrt{\log\frac{1}{\delta}} \right) \left( \frac{GC_g\sqrt{\log\frac{1}{\delta}} + \frac{G}{c_\eta}}{\epsilon^2 r^2} + \frac{\frac{\rho^2}{c_\eta} + \frac{G}{r}C_g\sqrt{\log\frac{1}{\delta}}}{\epsilon^2} \right).$$

To prove Theorem 3, we need the following lemma.

**Lemma 3.** *For Algorithm 1, the following statement holds true for any $\lambda \geq 0$ with probability at least $1 - \delta$:*

$$\left( f(\bar{\theta}_N) - f(\theta^*) \right) + \lambda\tilde{g}(\bar{\theta}_T) - \left( \frac{\gamma}{2} + \frac{1}{2\eta N} \right)\lambda^2$$

$$\leq \frac{\rho^2}{\eta N} + \eta C_g^2 + \eta G^2 + \frac{C_g\sqrt{3\log\frac{2}{\delta}}}{\sqrt{N}}(\lambda^* + \sqrt{2}\rho) + \frac{\sqrt{2}\rho(1 + \sqrt{2}\rho + \lambda^*)G\sqrt{3\log\frac{2}{\delta}}}{\sqrt{N}}$$

$$+ \epsilon_0(\lambda^* + \sqrt{2}\rho) + \left( \frac{C_g\sqrt{3\log\frac{2}{\delta}}}{\sqrt{N}} + \epsilon_0 \right)\lambda.$$

*Proof.* Define $L(\theta, \lambda) \doteq f(\theta) + \lambda \tilde{g}(\theta) - \frac{\gamma \lambda^2}{2}$, and let $\lambda^* = \arg\max_{\lambda \geq 0} \min f(\theta) + \lambda \tilde{g}(\theta)$. We first show that $\|\theta_t - \theta^*\|^2 + \|\lambda_t - \lambda^*\|^2 \leq 2\rho^2$, for any $t \in [T]$. We prove this by induction. Assume this holding for $t$, and for $t + 1$ we have

$$\|\theta_{t+1} - \theta^*\|^2 = \|\theta_t - \eta g_t - \theta^*\|^2$$
$$= \|\theta_t - \theta^*\|^2 - 2\langle \eta g_t, \theta_t - \theta^* \rangle + \eta^2 \|g_t\|^2,$$

where $g_t = \nabla f(\theta_t; \zeta_{f,t}) + \lambda_t \nabla g(\theta_t; \zeta_{g,t})$.

Then for $t + 1$, we have:

$$\|\theta_{t+1} - \theta^*\|^2 \leq \|\theta_t - \theta^*\|^2 - 2\eta_t \langle \nabla f(\theta_t) + \lambda_t \nabla g(\theta_t), \theta_t - \theta^* \rangle + 2\sqrt{2}\eta p_t \rho + \eta_t^2 (1 + \lambda_t)^2 G^2$$
$$\leq \|\theta_t - \theta^*\|^2 - 2\eta \left( L(\theta_t, \lambda_t) - L(\theta^*, \lambda_t) \right) + 2\sqrt{2}\eta p_t \rho + \eta^2 (1 + \lambda_t)^2 G^2$$

where $p_t = \|\nabla f(\theta_t) + \lambda_t \nabla g(\theta_t) - g_t\|$, and at last step we use the convexity of $L(\cdot, \lambda_t)$. We know that

$$\mathbb{E}[g_t] = \nabla f(\theta_t) + \lambda_t \nabla g(\theta_t), \mathbb{E}[\exp\left(p_t^2 / (G^2 + \lambda_t G^2)\right)] \leq \exp(1)$$

. Similarly, we have:

$$|\lambda_{t+1} - \lambda^*|^2 = |\lambda_t - \lambda^*|^2 + 2\eta \langle g(\theta_t; \zeta_{g,t}) - \gamma \lambda_t, \lambda_t - \lambda^* \rangle + \eta^2 |g(\theta_t; \zeta_{g,t}) - \gamma \lambda_t|^2$$
$$\leq (1 - \gamma \eta)|\lambda_t - \lambda|^2 - 2\eta \left( L(\theta_t, \lambda^*) - L(\theta_t, \lambda_t) \right)$$
$$+ 2\sqrt{2}\eta q_t \rho + 2\sqrt{2}\eta h_t \rho + 2\eta^2 C_g^2,$$

where

$$q_t = |g(\theta_t; \zeta_{g,t}) - g(\theta_t)|, h_t = |g(\theta_t) - \tilde{g}(\theta_t)| = \epsilon_0$$

and at last step we use the $\gamma$-strong-concavity of $L(\theta_t, \cdot)$. It is easy to see that

$$\mathbb{E}[g(\theta_t; \zeta_{g,t})] = g(\theta_t), \mathbb{E}[\exp\left(q_t^2 / C_g^2\right)] \leq \exp(1).$$

Putting pieces together we have

$$\|\theta_{t+1} - \theta^*\|^2 + |\lambda_{t+1} - \lambda^*|^2 \leq \left( |\lambda_t - \lambda^*|^2 + \|\theta_t - \theta^*\|^2 \right) - 2\eta \left( L(\theta_t, \lambda^*) - L(\theta^*, \lambda_t) \right)$$
$$+ 2\sqrt{2}\eta p_t \rho + 2\sqrt{2}\eta q_t \rho + 2\sqrt{2}\eta \epsilon \rho + 2\eta^2 C_g^2 + \eta^2 (1 + \lambda_t)^2 G^2$$
$$\leq \left( |\lambda_t - \lambda^*|^2 + \|\theta_t - \theta^*\|^2 \right) - 2\eta \underbrace{\left( L(\theta_t, \lambda^*) - L(\theta^*, \lambda_t) \right)}_{\geq -\frac{\gamma(\lambda^*)^2}{2}}$$
$$+ 2\eta^2 C_g^2 + \eta^2 (1 + \sqrt{2}\rho + \lambda^*)^2 G^2 + 2\sqrt{2}\eta \rho (p_t + q_t + \epsilon_0)$$

where the last step is due to

$$L(\theta_t, \lambda^*) - L(\theta^*, \lambda_t) = \underbrace{f(\theta_t) + \lambda^* \tilde{g}(\theta_t) - (f(\theta^*) + \lambda_t \tilde{g}(\theta^*))}_{\geq 0} - \frac{\gamma(\lambda^*)^2}{2} + \frac{\gamma \lambda_t^2}{2}.$$

Performing telescoping sum yields:

$$\|\theta_{t+1} - \theta^*\|^2 + |\lambda_{t+1} - \lambda^*|^2 \leq \left( \|\theta_0 - \theta^*\|^2 + |\lambda_0 - \lambda^*|^2 \right) + 2\eta^2 C_g^2 + \eta^2 (1 + \sqrt{2}\rho + \lambda^*)^2 G^2$$
$$+ 2\sqrt{2}\eta \rho \sum_{s=0}^t p_s + 2\sqrt{2}\eta \rho \sum_{s=0}^t q_s + 2\sqrt{2}\eta \rho t \epsilon_0 + N \gamma \eta (\lambda^*)^2.$$

Due to Lemma 4 of (Lan, 2012), we know with probability $1 - \delta/2$,

$$\sum_{t=0}^{N-1} p_t \leq (1 + \lambda_t) G \sqrt{N} \sqrt{3 \log \frac{2}{\delta}}, \sum_{t=0}^{N-1} q_t \leq \sqrt{N C_g^2} \sqrt{3 \log \frac{2}{\delta}}, \tag{28}$$

Putting pieces together yields:

$$
\|\theta_{t+1} - \theta^*\|^2 + |\lambda_{t+1} - \lambda^*|^2 \le |\lambda_0 - \lambda^*|^2 + \|\theta_0 - \theta^*\|^2 + 2\eta^2 C_g^2 + \eta^2(1 + \sqrt{2}\rho + \lambda^*)^2 G^2
$$
$$
+ 2\sqrt{2}\eta\rho\sqrt{N}(1 + \sqrt{2}\rho + \lambda^*)G\sqrt{3\log\frac{2}{\delta}}
$$
$$
+ 2\sqrt{2}\eta\rho\sqrt{N}C_g\sqrt{3\log\frac{2}{\delta}} + 2\sqrt{2}\eta\rho N\epsilon_0 + T\gamma\eta(\lambda^*)^2. \tag{29}
$$

Since we choose $\eta = \frac{c_\eta}{\sqrt{N}}$ and $\gamma = G^2\eta$, where

$$
c_\eta \le \min\left\{\frac{\rho}{2\sqrt{3}C_g}, \frac{\rho}{12\sqrt{6}(1 + \sqrt{2}\rho + \lambda^*)G\sqrt{3\log\frac{2}{\delta}}}, \frac{\rho}{12\sqrt{6}C_g\sqrt{\log\frac{2}{\delta}}}, \frac{\rho}{32\sqrt{N}\epsilon}\right\},
$$

we conclude that $\|\theta_{t+1} - \theta^*\|^2 + |\lambda_{t+1} - \lambda^*|^2 \le 2\rho^2$.

Now by similar analysis we have that for any $\lambda \ge 0$

$$
\|\theta_{t+1} - \theta^*\|^2 + |\lambda_{t+1} - \lambda|^2 \le \|\theta_t - \theta\|^2 + |\lambda_t - \lambda|^2 - 2\eta(L(\theta_t, \lambda) - L(\theta, \lambda_t))
$$
$$
+ 2\eta^2 C_g^2 + \eta^2(1 + \lambda_t)^2 G^2 + 2\eta\langle g(\theta_t; \zeta_{g,t}) - g(\theta_t), \lambda_t - \lambda\rangle
$$
$$
+ 2\eta\langle g(\theta_t) - \tilde{g}(\theta_t), \lambda_t - \lambda\rangle + 2\eta p_t \|\theta_t - \theta^*\|
$$
$$
\le \|\theta_t - \theta^*\|^2 + |\lambda_t - \lambda|^2 - 2\eta(L(\theta_t, \lambda) - L(\theta^*, \lambda_t))
$$
$$
+ 2\eta^2 C_g^2 + \eta^2(1 + \lambda_t)^2 G^2 + 2\eta q_t(\lambda_t + \lambda)
$$
$$
+ 2\eta\epsilon_0(\lambda_t + \lambda) + 2\sqrt{2}\eta p_t\rho.
$$

Since $|\lambda_t - \lambda^*| \le \sqrt{2}\rho$ we know $\lambda_t \le \lambda^* + \sqrt{2}\rho$. Hence we have

$$
\|\theta_{t+1} - \theta^*\|^2 + |\lambda_{t+1} - \lambda|^2 \le |\lambda_t - \lambda|^2 + \|\theta_t - \theta^*\|^2 - 2\eta(L(\theta_t, \lambda) - L(\theta^*, \lambda_t))
$$
$$
+ 2\eta^2 C_g^2 + \eta^2(1 + \lambda_t)^2 G^2 + 2\eta q_t\left(\lambda^* + \sqrt{2}\rho\right)
$$
$$
+ 2\eta\epsilon_0\left(\lambda^* + \sqrt{2}\rho\right) + 2\eta q_t\lambda + 2\eta\epsilon_0\lambda + 2\sqrt{2}\eta p_t\rho.
$$

Performing telescoping sum yields:

$$
\frac{1}{N}\sum_{t=0}^{N-1} L(\theta_t, \lambda) - L(\theta^*, \lambda_t) \le \frac{1}{2\eta N}(|\lambda_0 - \lambda|^2 + \|\theta_0 - \theta\|^2) + \frac{1}{N}\eta C_g^2 + \frac{1}{2N}\eta\sum_{t=0}^{N-1}(1 + \lambda_t)^2 G^2
$$
$$
+ \frac{1}{N}\sum_{t=0}^{N-1} q_t\left(\lambda^* + \sqrt{2}\rho\right) + \frac{1}{N}\sum_{t=0}^{N-1}\epsilon_0\left(\lambda^* + \sqrt{2}\rho\right) + \frac{1}{N}\sum_{t=0}^{N-1} q_t\lambda
$$
$$
+ \epsilon_0\lambda + \sqrt{2}\rho\frac{1}{N}\sum_{t=0}^{N-1} p_t.
$$

By the definition of Lagrangian, we have

$$
\frac{1}{N}\sum_{t=0}^{N-1}(f(\theta_t) + \lambda\tilde{g}(\theta_t) - \frac{\gamma}{2}\lambda^2 - f(\theta^*) - \lambda_t\underbrace{\tilde{g}(\theta^*)}_{\le 0} + \frac{\gamma}{2}\lambda_t^2)
$$
$$
\le \frac{1}{2\eta N}(|\lambda_0 - \lambda|^2 + \|\theta_0 - \theta^*\|^2) + \frac{1}{N}\eta C_g^2 + \frac{1}{2N}\eta\sum_{t=0}^{N-1}(1 + \lambda_t)^2 G^2
$$
$$
+ \frac{1}{N}\sum_{t=0}^{N-1} q_t\left(\lambda^* + \sqrt{2}\rho\right) + \epsilon_0\left(\lambda^* + \sqrt{2}\rho\right) + \frac{1}{N}\sum_{t=0}^{N-1} q_t\lambda + \epsilon_0\lambda + \sqrt{2}\rho\frac{1}{N}\sum_{t=0}^{N-1} p_t.
$$

Evoking the bound from (28) yields:

$$
\frac{1}{N} \sum_{t=0}^{N-1} (f(\theta_t) + \lambda \tilde{g}(\theta_t) - \frac{\gamma}{2}\lambda^2 - f(\theta^*) + \frac{\gamma}{2}\lambda_t^2)
$$

$$
\leq \frac{1}{2\eta N}(|\lambda_0 - \lambda|^2 + \|\theta_0 - \theta^*\|^2) + \frac{1}{N}\eta C_g^2 + \frac{1}{2N}\eta \sum_{t=0}^{N-1}(1+\lambda_t)^2 G^2
$$

$$
+ \frac{1}{N}\sqrt{TC_g^2}\sqrt{3\log\frac{2}{\delta}}\left(\lambda^* + 2\sqrt{2}\rho\right) + \epsilon_0\left(\lambda^* + \sqrt{2}\rho\right)
$$

$$
+ \frac{1}{N}\sqrt{TC_g^2}\sqrt{3\log\frac{2}{\delta}}\lambda + \epsilon_0\lambda + \frac{\sqrt{2}\rho(1+\sqrt{2}\rho+\lambda^*)G\sqrt{3\log\frac{2}{\delta}}}{\sqrt{N}}.
$$

Plugging in $\lambda_0 = 0$, $\theta_0 = \mathbf{0}$ and re-arranging the terms yields:

$$
\frac{1}{N}\sum_{t=0}^{N-1}(f(\theta_t) - f(\theta^*)) + \frac{1}{N}\sum_{t=0}^{N-1}\lambda\tilde{g}(\theta_t) - \left(\frac{\gamma}{2} + \frac{1}{2\eta N}\right)\lambda^2
$$

$$
\leq \frac{\rho^2}{\eta T} + \eta C_g^2 + \frac{1}{2N}\sum_{t=0}^{N-1}(\eta(1+\lambda_t)^2 G^2 - \gamma\lambda_t^2) + \frac{\sqrt{2}\rho(1+\sqrt{2}\rho+\lambda^*)G\sqrt{3\log\frac{2}{\delta}}}{\sqrt{N}}
$$

$$
+ \frac{C_g\sqrt{3\log\frac{2}{\delta}}}{\sqrt{N}}(\lambda^* + 2\sqrt{2}\rho) + \epsilon_0(\lambda^* + 2\sqrt{2}\rho) + \left(\frac{C_g\sqrt{3\log\frac{2}{\delta}}}{\sqrt{N}} + \epsilon_0\right)\lambda.
$$

By our choice, $\gamma = G^2\eta$, so we have

$$
\frac{1}{N}\sum_{t=0}^{N-1}(f(\theta_t) - f(\theta^*)) + \frac{1}{N}\sum_{t=0}^{N-1}\lambda\tilde{g}(\theta_t) - \left(\frac{\gamma}{2} + \frac{1}{2\eta N}\right)\lambda^2
$$

$$
\leq \frac{\rho^2}{\eta T} + \eta C_g^2 + \eta G^2 + \frac{C_g\sqrt{3\log\frac{2}{\delta}}}{\sqrt{N}}(\lambda^* + \sqrt{2}\rho) + \frac{\sqrt{2}\rho(1+\sqrt{2}\rho+\lambda^*)G\sqrt{3\log\frac{2}{\delta}}}{\sqrt{N}}
$$

$$
+ \epsilon_0(\lambda^* + \sqrt{2}\rho) + \left(\frac{C_g\sqrt{3\log\frac{2}{\delta}}}{\sqrt{N}} + \epsilon_0\right)\lambda.
$$

Define $\hat{\theta}_T = \frac{1}{N}\sum_{t=0}^{N-1}\theta_T$, and then by Jensen's inequality we have

$$
\left(f(\bar{\theta}_T) - f(\theta^*)\right) + \lambda\tilde{g}(\bar{\theta}_T) - \left(\frac{\gamma}{2} + \frac{1}{2\eta N}\right)\lambda^2
$$

$$
\leq \frac{\rho^2}{\eta T} + \eta C_g^2 + \eta G^2 + \frac{C_g\sqrt{3\log\frac{2}{\delta}}}{\sqrt{N}}(\lambda^* + \sqrt{2}\rho) + \frac{\sqrt{2}\rho(1+\sqrt{2}\rho+\lambda^*)G\sqrt{3\log\frac{2}{\delta}}}{\sqrt{N}}
$$

$$
+ \epsilon_0(\lambda^* + \sqrt{2}\rho) + \left(\frac{C_g\sqrt{3\log\frac{2}{\delta}}}{\sqrt{N}} + \epsilon_0\right)\lambda.
$$

$\square$

## C.1 PROOF OF THEOREM 3

*Proof.* Note that Lemma 3 holds for any $\lambda \geq 0$. Now let's discuss by cases. If $\bar{\theta}_T$ is in the constraint set, then $\hat{\theta} = \bar{\theta}_T$ and we simply set $\lambda = 0$ and get the convergence:

$$
\left(f(\bar{\theta}_T) - f(\theta^*)\right) + \lambda \tilde{g}(\bar{\theta}_T) - \left(\frac{\gamma}{2} + \frac{1}{2\eta N}\right)\lambda^2
$$

$$
\leq \frac{\rho^2}{\eta T} + \eta C_g^2 + \eta G^2 + \frac{C_g\sqrt{3\log\frac{2}{\delta}}}{\sqrt{N}}(\lambda^* + \sqrt{2}\rho) + \frac{\sqrt{2}\rho(1 + \sqrt{2}\rho + \lambda^*)G\sqrt{3\log\frac{2}{\delta}}}{\sqrt{N}}
$$

$$
+ \epsilon(\lambda^* + \sqrt{2}\rho) + \left(\frac{C_g\sqrt{3\log\frac{2}{\delta}}}{\sqrt{N}} + \epsilon\right)\lambda.
$$

If $\bar{\theta}_T$ is not in the constraint set, we set $\lambda = \frac{\tilde{g}(\bar{\theta}_T)}{\gamma + \frac{1}{\eta T}}$, which yields:

$$
(f(\bar{\theta}_T) - f(\theta^*)) + \frac{(\tilde{g}(\bar{\theta}_T))^2}{2(\gamma + \frac{1}{\eta T})}
$$

$$
\leq \frac{\rho^2}{\eta T} + \eta C_g^2 + \eta G^2 + \frac{C_g\sqrt{3\log\frac{2}{\delta}}}{\sqrt{N}}\left(\lambda^* + 2\sqrt{2}\rho\right) + \frac{\sqrt{2}\rho(1 + \sqrt{2}\rho + \lambda^*)G\sqrt{3\log\frac{2}{\delta}}}{\sqrt{N}}
$$

$$
+ \epsilon_0\left(\lambda^* + \sqrt{2}\rho\right) + \left(\frac{C_g\sqrt{3\log\frac{2}{\delta}}}{\sqrt{N}} + \epsilon\right)\left|\frac{\tilde{g}(\bar{\theta}_T)}{\gamma + \frac{1}{\eta T}}\right|. \tag{30}
$$

Since $\bar{\theta}_T$ is not in the constraint set and $\hat{\theta}$ is the projection of it onto inexact constraint set $\hat{g}(\theta) \leq -\delta$, by KKT condition we know $g(\hat{\theta}) = -\delta$ and $\bar{\theta}_T - \hat{\theta} = s \cdot \nabla g(\hat{\theta})$ for some $s > 0$ where $g'(\theta) \doteq g(\theta) + \delta$. Defining $\Delta := \delta - c$, and due to our choice of $T$ we know $\Delta \geq 0$. Then we have

$$
\begin{aligned}
\tilde{g}(\bar{\theta}_T) &= \tilde{g}(\bar{\theta}_T) - g'(\hat{\theta}) \\
&= \tilde{g}(\bar{\theta}_T) - \tilde{g}(\hat{\theta}) - (g'(\bar{\theta}_T) - \tilde{g}(\bar{\theta}_T)) \\
&= \tilde{g}(\bar{\theta}_T) - \tilde{g}(\hat{\theta}) - (\delta - c) \\
&\geq \langle\nabla\tilde{g}(\hat{\theta}), \bar{\theta}_T - \hat{\theta}\rangle - \Delta = \left\|\nabla\tilde{g}(\hat{\theta})\right\|\left\|\bar{\theta}_T - \hat{\theta}\right\| - \Delta
\end{aligned}
$$

where the inequality is due to convexity of $\tilde{g}(\cdot)$. Let $\theta_0$ be such that $\tilde{g}(\theta_0) = 0$, and then we have

$$
\begin{aligned}
\min_{g'(\theta)=0}\|\nabla\tilde{g}(\theta)\| &\geq \min_{g'(\theta)=0}\|\nabla\tilde{g}(\theta_0)\| - \|\nabla\tilde{g}(\theta_0) - \nabla\tilde{g}(\theta)\| \\
&\geq r - 2L\left(\tilde{g}(\theta_0) - \tilde{g}(\theta)\right) = r - 2L\tilde{g}(\theta) \\
&= r - 2L(c - \delta) \\
&\geq r - 2L\epsilon_0
\end{aligned}
$$

, so $g(\bar{\theta}_T) \geq (r - 2L\epsilon_0)\left\|\bar{\theta}_T - \hat{\theta}\right\| - \Delta$. On the other hand, since $\hat{\theta}$ is the projection of $\bar{\theta}_T$ onto constraint set, and $\theta^*$ is in the constraint set, we know

$$
\left\|\hat{\theta} - \theta^*\right\|^2 \leq \left\|\bar{\theta}_T - \theta^*\right\|^2 \leq 2\rho^2.
$$

Hence we also know

$$
\begin{aligned}
\tilde{g}(\bar{\theta}_T) &= \tilde{g}(\bar{\theta}_T) - g'(\hat{\theta}) \\
&= \tilde{g}(\bar{\theta}_T) - g(\hat{\theta}) - \delta \\
&= \tilde{g}(\bar{\theta}_T) - \tilde{g}(\hat{\theta}) + \tilde{g}(\hat{\theta}) - g(\hat{\theta}) - \delta \\
&\leq G\left\|\bar{\theta}_T - \hat{\theta}\right\| + \underbrace{c - \delta}_{\leq 0}.
\end{aligned}
$$

Plugging the upper and lower bound of $g(\bar{\theta}_T)$ into (30) yields:

$$(f(\bar{\theta}_T) - f(\theta^*)) + \sqrt{N}\frac{((r - 2H\epsilon_0)\|\bar{\theta}_T - \hat{\theta}\| - \Delta)^2}{2(c_\eta G^2 + \frac{1}{c_\eta})}$$

$$\leq \frac{\rho^2}{\eta T} + \eta C_g^2 + \eta G^2 + \frac{C_g\sqrt{3\log\frac{2}{\delta}}}{\sqrt{N}}\left(\lambda^* + 2\sqrt{2}\rho\right) + \frac{\sqrt{2}\rho(1 + \sqrt{2}\rho + \lambda^*)G\sqrt{3\log\frac{2}{\delta}}}{\sqrt{N}}$$

$$+ \epsilon_0\left(\lambda^* + \sqrt{2}\rho\right) + \left(\frac{C_g\sqrt{3\log\frac{2}{\delta}}}{\sqrt{N}} + \epsilon_0\right)\frac{\sqrt{N}}{2(c_\eta G^2 + \frac{1}{c_\eta})}\hat{G}\left\|\bar{\theta}_T - \hat{\theta}\right\|. \tag{31}$$

Notice the following decomposition: $f(\bar{\theta}_T) - f(\theta^*) \geq f(\bar{\theta}_T) - f(\hat{\theta}) \geq -\hat{G}\left\|\bar{\theta}_T - \hat{\theta}\right\|$. Also notice the fact $(a - b)^2 \geq \frac{1}{2}a^2 - b^2$ holding for any $a > 0, b > 0$, we know

$$\left((r - 2H\epsilon_0)\left\|\bar{\theta}_T - \hat{\theta}\right\| - \Delta\right)^2 \geq \frac{1}{2}(r - 2H\epsilon_0)^2\left\|\bar{\theta}_T - \hat{\theta}\right\|^2 - \Delta^2.$$

Putting pieces together yields the following inequality:

$$a\left\|\bar{\theta}_T - \hat{\theta}\right\|^2 - b\left\|\bar{\theta}_T - \hat{\theta}\right\| - c \leq 0,$$

where:

$$a = \frac{\sqrt{N}(r - 2H\epsilon_0)^2}{4(c_\eta G^2 + \frac{1}{c_\eta})}$$

$$b = \frac{G}{(c_\eta G^2 + \frac{1}{c_\eta})}\left(C_g\sqrt{3\log\frac{2}{\delta}} + \sqrt{N}\epsilon_0\right) + G,$$

$$c = \frac{\frac{\rho^2}{c_\eta} + c_\eta(C_g^2 + G^2) + C_g\sqrt{3\log\frac{2}{\delta}}\left(\lambda^* + 2\sqrt{2}\rho\right) + \sqrt{2}\rho(1 + \sqrt{2}\rho + \lambda^*)G\sqrt{3\log\frac{2}{\delta}}}{\sqrt{N}}$$

$$+ \epsilon_0\left(\lambda^* + \sqrt{2}\rho\right) + \sqrt{N}\frac{\Delta^2}{2(c_\eta G^2 + \frac{1}{c_\eta})}.$$

Assume $\epsilon_0 \leq \frac{C_g^2 3\log\frac{2}{\delta}}{\sqrt{N}}$, so $b \leq \frac{2GC_g\sqrt{3\log\frac{2}{\delta}}}{(c_\eta G^2 + \frac{1}{c_\eta})} + G$. Solving the above quadratic inequality yields:

$$\left\|\bar{\theta}_T - \hat{\theta}\right\| \leq \frac{b + \sqrt{b^2 + 4ac}}{2a} \leq \frac{b}{a} + \sqrt{\frac{c}{a}}$$

$$\leq \frac{2GC_g\sqrt{3\log\frac{2}{\delta}}}{\sqrt{N}(r - 2H\epsilon_0)^2} + \frac{4G(c_\eta G^2 + \frac{1}{c_\eta})}{\sqrt{N}(r - 2H\epsilon_0)^2} + \frac{\Delta}{2(r - 2H\epsilon_0)}$$

$$+ \sqrt{\frac{4(c_\eta G^2 + \frac{1}{c_\eta})}{\sqrt{N}(r - 2H\epsilon_0)^2}}\sqrt{\frac{\frac{\rho^2}{c_\eta} + c_\eta(C_g^2 + G^2) + (C_g\left(\lambda^* + 2\sqrt{2}\rho\right) + \sqrt{2}\rho(1 + \sqrt{2}\rho + \lambda^*)G)\sqrt{3\log\frac{2}{\delta}}}{\sqrt{N}}}$$

$$+ \sqrt{\frac{4(c_\eta G^2 + \frac{1}{c_\eta})}{\sqrt{N}(r - 2H\epsilon_0)^2}}\epsilon_0\left(\lambda^* + \sqrt{2}\rho\right)$$

$$= \frac{2GC_g\sqrt{3\log\frac{2}{\delta}} + 6G(c_\eta G^2 + \frac{1}{c_\eta})}{\sqrt{N}(r - 2H\epsilon_0)^2} + \frac{\Delta}{2(r - 2H\epsilon_0)}$$

$$+ \frac{\frac{\rho^2}{c_\eta} + c_\eta(C_g^2 + G^2) + C_g\sqrt{3\log\frac{2}{\delta}}\left(\lambda^* + 2\sqrt{2}\rho\right) + \sqrt{2}\rho(1 + \sqrt{2}\rho + \lambda^*)G\sqrt{3\log\frac{2}{\delta}}}{2\sqrt{N}} + \epsilon_0\left(\lambda^* + \sqrt{2}\rho\right)$$

where at the last step we use the fact $\sqrt{ab} \leq \frac{a^2+b^2}{2}$. Finally, note the following decomposition:

$$
\begin{aligned}
f(\hat{\theta}) - f(\theta^*) &= f(\hat{\theta}) - f(\bar{\theta}_T) + f(\bar{\theta}_T) - f(\theta^*) \\
&\leq G \left\| \bar{\theta}_T - \hat{\theta} \right\| + f(\bar{\theta}_T) - f(\theta^*) \\
&\leq \left( G + \frac{2C_g\sqrt{3\log\frac{2}{\delta}}}{c_\eta G^2 + \frac{1}{c_\eta}} G \right) \left\| \bar{\theta}_T - \hat{\theta} \right\| + \epsilon_0 \left( \lambda^* + \sqrt{2}\rho \right) \\
&+ \frac{\frac{\rho^2}{c_\eta} + c_\eta(C_g^2 + G^2) + (C_g\left(\lambda^* + 2\sqrt{2}\rho\right) + \sqrt{2}\rho(1 + \sqrt{2}\rho + \lambda^*)G)\sqrt{3\log\frac{2}{\delta}}}{\sqrt{N}}.
\end{aligned}
$$

Recalling $\epsilon_0 \leq \frac{C_g\sqrt{3\log\frac{2}{\delta}}}{(\lambda^*+\sqrt{2}\rho)\sqrt{N}}$ and plugging bound of $\left\| \bar{\theta}_T - \hat{\theta} \right\|$ yields:

$$
\begin{aligned}
&f(\hat{\theta}) - f(\theta^*) \\
&\leq \left( G + \frac{2C_g\sqrt{3\log\frac{2}{\delta}}}{c_\eta G^2 + \frac{1}{c_\eta}} G \right) \left( \frac{2GC_g\sqrt{3\log\frac{2}{\delta}} + 6G(c_\eta G^2 + \frac{1}{c_\eta})}{\sqrt{N}(r - 2H\epsilon_0)^2} + \frac{\Delta}{2(r - 2H\epsilon_0)} \right) \\
&+ \left( 1 + G + \frac{C_g\sqrt{\log\frac{2}{\delta}}}{c_\eta G^2 + \frac{1}{c_\eta}} G \right) \left( \frac{\frac{\rho^2}{c_\eta} + c_\eta(C_g^2 + G^2) + 2C_g\sqrt{\log\frac{2}{\delta}}\left(\lambda^* + \rho\right) + \rho G\sqrt{\log\frac{2}{\delta}}}{\sqrt{N}} \right).
\end{aligned}
$$

Now we simplify the above bound. By the definition of $c_\eta$ we know $c_\eta \leq \frac{1}{G}$, so we have

$$
\begin{aligned}
&f(\hat{\theta}) - f(\theta^*) \\
&\leq \left( 2G + \frac{2C_g\sqrt{3\log\frac{2}{\delta}}}{c_\eta G + 1} \right) \left( \frac{2GC_g\sqrt{3\log\frac{2}{\delta}} + 6G(c_\eta G^2 + \frac{1}{c_\eta})}{\sqrt{N}(r - 2H\epsilon_0)^2} + \frac{\Delta}{2(r - 2H\epsilon_0)} \right) \\
&+ \left( 2G + \frac{2C_g\sqrt{3\log\frac{2}{\delta}}}{c_\eta G + 1} \right) \left( \frac{\frac{\rho^2}{c_\eta} + c_\eta(C_g^2 + G^2) + (C_g\left(\lambda^* + \rho\right) + \rho G)\sqrt{\log\frac{2}{\delta}}}{\sqrt{N}} \right).
\end{aligned}
$$

Finally, by definition of $c_\eta$ we know $\frac{\rho^2}{c_\eta} \geq c_\eta C_g^2$, $\frac{\rho^2}{c_\eta} \geq \rho C_g$, $\frac{\rho^2}{c_\eta} \geq \rho 12\sqrt{6}G\sqrt{\log\frac{2}{\delta}}$ and $\frac{G/c_\eta}{(r-2H\epsilon_0)^2} \geq c_\eta G^2$ which concludes the proof:

$$
f(\hat{\theta}) - f(\theta^*) \lesssim \left( G + C_g\sqrt{\log\frac{1}{\delta}} \right) \left( \frac{GC_g\sqrt{\log\frac{1}{\delta}} + \frac{G}{c_\eta}}{\sqrt{N}(r - 2H\epsilon_0)^2} + \frac{\Delta}{(r - 2H\epsilon_0)} + \frac{\frac{\rho^2}{c_\eta} + \lambda^*C_g\sqrt{\log\frac{1}{\delta}}}{\sqrt{N}} \right)
$$

Plugging the bound that $\lambda^* \leq G/r$ (see (Mahdavi et al., 2012, Remark 1)) will conclude the proof. $\square$

### C.2 PROOF OF THEOREM 2

*Proof.* Now we proceed to proving Theorem 2. We split the proof into two parts: proof of optimization guarantee (the gradient complexity) and proof of statistical rate.

**Convergence rate** The total gradient complexity of Algorithm 2 will be the sum of the complexities of each calling of procedure CP-Solver. Theorem 3 gives the complexity of CP-Solver, and

what we need to compute is $\epsilon_0$ in each call. Recall that, $\epsilon_0$ is the error between the true constraint and the approximate we actually used in `CP-Solver`. Hence, we first compute those error as follows:

$$|\hat{g}'(\theta, \alpha') - \hat{g}'(\theta, \alpha')| \leq |\hat{g}_{1,T}(\theta) - g_{1,T}(\theta)| \leq \hat{R}_{\varphi,\mu_{1,T}}(\hat{\theta}_{T,\alpha-\epsilon_{0,T}}) - \hat{R}_{\varphi,\mu_{1,T}}(\hat{\theta}^*_{T,\alpha-\epsilon_{0,T}})$$

$$\leq \epsilon_{T,\alpha-\epsilon_{0,T}},$$

$$|\hat{g}_{\hat{\alpha}}(\theta) - g_{\hat{\alpha}_S}(\theta)| \leq \max\left\{|\hat{\alpha} - \hat{\alpha}_S|, |\hat{g}_{1,T}(\theta) - g_{1,T}(\theta)|\right\} \leq \max\left\{\epsilon_{\hat{\alpha}_S}, \epsilon_{T,\alpha-\epsilon_{0,T}}\right\},$$

$$|\hat{g}_{S,T}(\theta) - g_{S,T}(\theta)| \leq \max\left\{|\hat{g}_{\hat{\alpha}}(\theta) - g_{\hat{\alpha}_S}(\theta)|, |\hat{g}'_T(\theta) - g'_T(\theta)|\right\}$$

$$\leq \max\left\{\epsilon_{\hat{\alpha}_S}, \epsilon_{T,\alpha-\epsilon_{0,T}}, \hat{R}_{1,T}(\hat{\theta}^*_{T,\hat{\alpha}_S}) - \hat{R}_{1,T}(\hat{\theta}_{T,\hat{\alpha}_S})\right\}$$

$$\leq \max\left\{\epsilon_{\hat{\alpha}_S}, \epsilon_{T,\alpha-\epsilon_{0,T}}, \epsilon'_T\right\}.$$

Now we verify that the choice of the tolerance in Algorithm 2 can ensure that

$$\hat{R}_{\varphi,\mu_{1,S}}(\tilde{\theta}) - \min_{\theta:g_{S,T}(\theta)\leq 0} \hat{R}_{\varphi,\mu_{1,S}}(\theta) \leq \epsilon_{1,S} = \epsilon_{S,T}.$$

To ensure `CP-Solver`$\left(\hat{R}_{\varphi,\mu_{1,S}}, \hat{g}_{S,T}, \xi(\epsilon_{S,T}, r_{S,T}), \epsilon_{S,T}\right)$ outputs $\epsilon_{S,T}$-accurate solution, by the condition in Theorem 3, we need $|\hat{g}_{S,T}(\theta) - g_{S,T}(\theta)| \leq \xi(\epsilon_{S,T}, r_{S,T})$, hence we require

$$\epsilon_{\hat{\alpha}_S} \leq \xi(\epsilon_{S,T}, r_{S,T}), \tag{32}$$

$$\epsilon_{T,\alpha-\epsilon_{0,T}} \leq \xi(\epsilon_{S,T}, r_{S,T}), \tag{33}$$

$$\epsilon'_T \leq \xi(\epsilon_{S,T}, r_{S,T}) \tag{34}$$

We also require $\epsilon'_T \leq \epsilon_{1,T}$, so we know choosing $\epsilon'_T = \min\{\xi(\epsilon_{S,T}, r_{S,T}), \epsilon_{1,T}\}$ suffices.

To ensure `CP-Solver`$\left(\hat{R}_{\varphi,\mu_{1,T}}, \hat{g}_{\hat{\alpha}}, \xi(\epsilon'_T, r'_T), \epsilon'_T\right)$ output $\epsilon'_T$-accurate solution, we need $|\hat{g}_{\hat{\alpha}}(\theta) - g_{\hat{\alpha}_S}(\theta)| \leq \xi(\epsilon'_T, r'_T)$, hence we require

$$\epsilon_{\hat{\alpha}_S} \leq \xi(\epsilon'_T, r'_T), \tag{35}$$

$$\epsilon_{T,\alpha-\epsilon_{0,T}} \leq \xi(\epsilon'_T, r'_T) \tag{36}$$

To ensure `CP-Solver`$\left(\hat{R}_{\varphi,\mu_{1,S}}, \hat{g}_{\hat{\alpha}}, \xi(\epsilon'_S, r'_S), \epsilon'_S\right)$ output $\epsilon'_S = \epsilon_{1,S}$-accurate solution, we need $|\hat{g}_{\hat{\alpha}}(\theta) - g_{\hat{\alpha}_S}(\theta)| \leq \xi(\epsilon'_S, r'_S)$, hence we require

$$\epsilon_{\hat{\alpha}_S} \leq \xi(\epsilon'_S, r'_S), \tag{37}$$

$$\epsilon_{T,\alpha-\epsilon_{0,T}} \leq \xi(\epsilon'_S, r'_S) \tag{38}$$

To ensure `CP-Solver`$(\alpha', \hat{g}'(\theta, \alpha'), \xi(\epsilon_{\hat{\alpha}_S}, r_{\hat{\alpha}_S}), \epsilon_{\hat{\alpha}_S})$ output $\epsilon_{\hat{\alpha}_S}$-accurate solution, we need $|\hat{g}_{1,T}(\theta) - g_{1,T}(\theta)| \leq \xi(\epsilon_{\hat{\alpha}_S}, r_{\hat{\alpha}_S})$, hence we require

$$\epsilon_{T,\alpha-\epsilon_{0,T}} \leq \xi(\epsilon_{\hat{\alpha}_S}, r_{\hat{\alpha}_S}) \tag{39}$$

From (32), (35) and (37) we know that choosing $\epsilon_{\hat{\alpha}_S} \leq \min\left\{\xi(\epsilon'_{S,T}, r'_{S,T}), \xi(\epsilon'_T, r'_T), \xi(\epsilon'_S, r'_S)\right\}$ suffices and from (33), (36), (37) and (39) we know that choosing $\epsilon_{T,\alpha-\epsilon_{0,T}} \leq \min\left\{\xi(\epsilon'_{S,T}, r'_{S,T}), \xi(\epsilon'_T, r'_T), \xi(\epsilon'_S, r'_S), \xi(\epsilon_{\hat{\alpha}_S}, r_{\hat{\alpha}_S})\right\}$ suffices.

The complexity immediately follows by plugging error tolerances into Theorem 3.

**Statistical Rate** If $\hat{R}_{\varphi,\mu_{1,S}}(\tilde{\theta}) - \hat{R}_{\varphi,\mu_{1,S}}(\hat{\theta}'_S) > 2\epsilon_{1,S}$, then we know

$$\min_{\theta:g_{S,T}(\theta)\leq 0} \hat{R}_{\varphi,\mu_{1,S}}(\theta) - \min_{\theta:g_{\hat{\alpha}_S}\leq\epsilon_{1,S}} \hat{R}_{\varphi,\mu_{1,S}}(\theta) \geq \hat{R}_{\varphi,\mu_{1,S}}(\tilde{\theta}) - \hat{R}_{\varphi,\mu_{1,S}}(\hat{\theta}'_S) - \epsilon_{1,S} \geq \epsilon_{1,S}$$

Hence the set $\{\theta : g_{\hat{\alpha}_S}(\theta) \leq 0, g'_T(\theta) \leq 0\}$ does not intersect with $\{\theta : g_{\hat{\alpha}_S}(\theta) \leq 0, g'_S(\theta) \leq \epsilon_{1,S}\}$ which is $\{h \in \hat{\mathcal{H}}' : \hat{R}_{\varphi,\mu_{1,S}}(h) \leq \hat{R}^*_{\varphi,\mu_{1,S}}(\hat{\mathcal{H}}') + 3\epsilon_{1,S}\}$ (a slightly inflated version of $\hat{\mathcal{H}}'_{1,S}$), so we output $\hat{\theta}'_T$.

Otherwise, $\hat{R}_{\varphi,\mu_{1,S}}(\hat{\theta}) - \hat{R}_{\varphi,\mu_{1,S}}(\hat{\theta}'_S) \leq 2\epsilon_{1,S}$, then

$$\min_{\theta: g_{S,T}(\theta) \leq 0} \hat{R}_{\varphi,\mu_{1,S}}(\theta) - \min_{\theta: g_{\hat{\alpha}_S} \leq \epsilon_{1,S}} \hat{R}_{\varphi,\mu_{1,S}}(\theta) \leq \hat{R}_{\varphi,\mu_{1,S}}(\tilde{\theta}) - \hat{R}_{\varphi,\mu_{1,S}}(\hat{\theta}'_S) - \epsilon_{1,S} \leq \epsilon_{1,S},$$

which implies that the set $\{\theta : g_{\hat{\alpha}_S}(\theta) \leq 0, g'_T(\theta) \leq 0\}$ $(\hat{\mathcal{H}}'_T)$ intersects with $\{\theta : g_{\hat{\alpha}_S}(\theta) \leq 0, g'_S(\theta) \leq -\epsilon_{1,s}\}$ which is $\{h \in \hat{\mathcal{H}}' : \hat{R}_{\varphi,\mu_{1,S}}(h) \leq \hat{R}^*_{\varphi,\mu_{1,S}}(\hat{\mathcal{H}}') + \epsilon_{1,S}\}$ (a slightly shrinked version of $\hat{\mathcal{H}}'_{1,S}$ ), so we output a model $\tilde{\theta}$ in their intersection.

Hence the output of our optimization procedure is consistent with the learning algorithm (8), and the same generalization analysis applies here. □

## D ADDITIONAL EXPERIMENTS

In this section, we present additional experiments on synthetic Gaussian data in dimension 10. The source class-0 distribution is $N(-2, I)$, the target class-0 distribution is $N(0, I)$, and both source and target class-1 distributions are $N(1, I)$. In Figure 4, we fix $n_{0,T} = n_{1,T} = 40$ and vary the number of source samples $n_{0,S} = n_{1,S}$ from 50 to 3000, with $\alpha = 0.1$. In Figure 5, we fix $n_{0,S} = n_{1,S} = 1000$ and vary the number of target samples $n_{0,T} = n_{1,T}$ from 50 to 1000, again with $\alpha = 0.1$. Each experiment is repeated over 10 independent trials.

The results in Figure 4 show that the source-only method exhibits a very large Type-I error, while the proposed method interpolates toward the target-only method as expected. It is important to note that, in the implementation, several constants appear in the quantities $\epsilon_{i,D}$ for $i \in \{0, 1\}$ and $D \in \{S, T\}$. These $\epsilon$-terms arise from uniform concentration inequalities controlling the gap between empirical and population errors. In theory, such inequalities are known to be conservative, and the associated constants are not tight; in practice, however, these parameters can be tuned more carefully. Consequently, while the proposed method provides practical insight into how to design an adaptive transfer-learning procedure, further tuning and experimental investigation are needed to select these constants appropriately, as they directly influence the final error performance.

In Figure 5, as the number of target samples increases, the type-I errors of both TLA and the target-only method converge to $\alpha = 0.1$. The slight increase in the type-II error of TLA reflects the inherent trade-off between type-I and type-II errors: reducing one typically leads to an increase in the other.

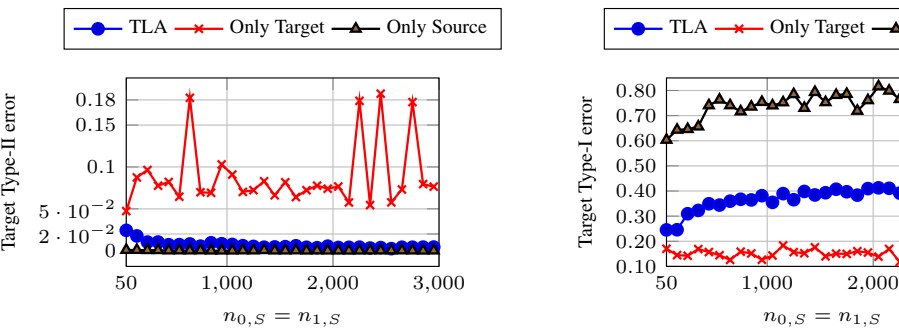

Figure 4: The performance of our algorithm (TLA), along with two baselines—using only source data and only target data—is evaluated under a Type-I error threshold of $\alpha = 0.1$. The experiment is conducted on synthetic Gaussian data in dimension 10, where the source class-0 distribution is $N(-2, I)$, the target class-0 distribution is $N(0, I)$, and both source and target class-1 distributions are $N(1, I)$. We fix $n_{0,T} = n_{1,T} = 40$ and vary the number of source samples $n_{0,S} = n_{1,S}$.

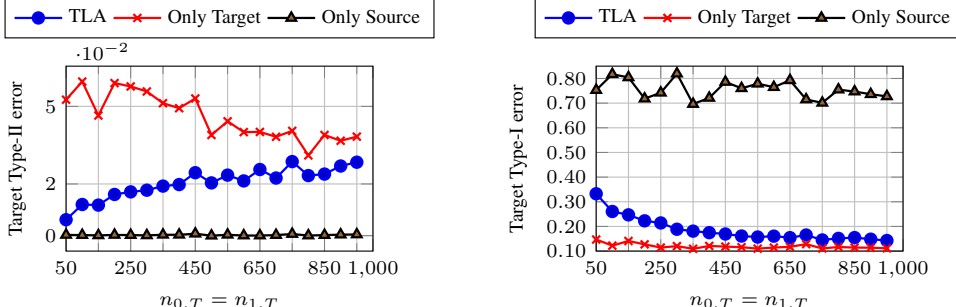

Figure 5: The performance of our algorithm (TLA), along with two baselines—using only source data and only target data—is evaluated under a Type-I error threshold of $\alpha = 0.1$. The experiment is conducted on synthetic Gaussian data in dimension 10, where the source class-0 distribution is $N(-2, I)$, the target class-0 distribution is $N(0, I)$, and both source and target class-1 distributions are $N(1, I)$. We fix $n_{0,S} = n_{1,S} = 1000$ and vary the number of Target samples $n_{0,T} = n_{1,T}$.

