# OpenReview forum: "Neyman-Pearson Classification under Both Null and Alternative Distributions Shift"
_ICLR.cc/2026/Conference — ICLR 2026 Poster_

### Official Review · Reviewer_dxrU · 2025-10-17

**Soundness:** 4
**Presentation:** 3
**Contribution:** 3
**Rating:** 8
**Confidence:** 3

**Summary:**

This paper provides an algorithm, to perform NP classification under distribution in the case of transfer learning where for both class 0 and 1 data there is a small amount of target data from our distribution of interest and a larger amount of source data from an unknown but potentially similar distribution. The NP classifier is shown to satisfy the Type I error constraints with high probability up to some error term $\epsilon$ (while at least matching the error bound given by just using Type I error) as well at least matching the Type II error given by a classifier just produced on the target data and improving upon it when the source and target are similar.

At a high level this procedure works by finding optimal classifiers on both the source and target data (with a relaxation of the Type I error on the source data to ensure some intersection exists) then takes the final classifier to be the intersection of the near optimal classifiers on both source and target data within this set (if such a classifier exists.)

**Strengths:**

* This work effectively extends the work of Kalan \& Kpotufe (2024); Kalan et al. (2025) to the case of distribution shift on the class 0 data. Furthermore the theoretical results illustrate meaningful improvement from the inclusion of source the data from both classes provided the transfer moduli is sufficiently small.

* Strong theoretical results are provided both for the optimal classifier under the schema as well as a classifier given by a practical algorithm solving the optimisation problem.

* The relative error induced by the difference in source and target distributions is nicely related to the transfer exponent, a pre-existing measure of distribution shift.

* Experimental results on real world data illustrate improvement when the source and target data overlap and minimal downside when the target and source data do not.

**Weaknesses:**

* The additional error on the target Type I error $\epsilon_{0,T}$ depends upon the Rademacher complexity $B_{\mathcal{H}}$ a quantity which while bounded for may function classes is not often practically known. This makes it difficult to use these results to give meaningful finite sample, high probability bounds on the Type I error for a specific case. This is in contrast to works such as Tong (2013) and Tong et al. (2018) which use a sample splitting procedure to project the problem down to learning the threshold of a score function, effectively making $\mathcal{H}$ have VC dimension 1.

* The paper lacks a proper conclusion or discussion of future directions/limitations of the work which I feel would enhance the paper, especially is it is a very dense technical work.

* The experimental results do not illustrate a case where improper handling of the source data harms Type I error, in both cases the lowest Type I error (theoretically the value we are most keen to control) are mostly given by exclusively working with the source data.

* While I appreciate that it is just a very technical work. I feel it could be improved by trying to streamline notation. E.g. later in the paper, the sets $\mathcal{H}', \mathcal{H}'_1, \mathcal{H}'_0$ are re-written as constraints on the functions $g$. Could these functions instead be introduced earlier therefore requiring fewer subsets of $\mathcal{H}$ to be defined?

* Some small errata (I think):
    * Line 343 -- Definition of $\hat{\theta}_{T,\alpha}$ not defined and I believe it should be $-6\epsilon_{1,T}$ is the definition of $g_{1,T}$.
    * Line 361 -- Typo, ``uniformly'' written twice.
    * Line 356 -- Slackness should be $\xi$? $\delta$ not used and neither is $\epsilon$ (though by CP-Solver's later use $\epsilon$ should appear.)
    * Line 420 -- Technically, (12) only finds $\hat{h}$ when $\mathcal{H}'_1\cap\mathcal{H}'_0\neq\varnothing$ and I feel this would be clearer if it was clarified (I appreciate that this is clarified in the algorithm but would benefit from clarification here as well.)

## References
Xin Tong. A plug-in approach to neyman-pearson classification. *The Journal of Machine Learning
Research*, 14(1):3011–3040, 2013.

Xin Tong, Yang Feng, and Jingyi Jessica Li. “Neyman-Pearson Classification
Algorithms and NP Receiver Operating Characteristics”. In: *Science Advances 4.2*
(Feb. 2, 2018),

**Questions:**

Can this be extended to the case where some relationship between the target and source distributions are known and can be leveraged for example the density ratio or some other relationship?

Are there illustrative examples of how the transfer modulus or transfer exponent vary as the source and target distributions differ from one another?

---

> ### Author Response · Authors · 2025-11-22
>
> Thank you for your review and feedback.
>
> **The additional error on the target Type I error depends upon the Rademacher complexity, a quantity...**
>
> Yes, we agree with the reviewer that for many hypothesis classes the exact Rademacher complexity is not known, although for some classes—such as linear models—it is well understood, and for neural networks there exist upper bounds [1]. However, this is a standard setting in learning theory.
>
> Regarding [2], the setting of that paper is different: it considers a nonparametric plug-in approach whose rates depend on the data dimension and therefore suffer from the curse of dimensionality, while our work is aligned with modern ML approaches, including deep neural networks. Furthermore, [3] considers a particular hypothesis class that first learns a scoring function and then tunes a threshold, whereas our framework considers a more general class.
>
> [1] Golowich, Noah, Alexander Rakhlin, and Ohad Shamir. "Size-independent sample complexity of neural networks." Conference On Learning Theory. 2018
>
> [2] Xin Tong. A plug-in approach to neyman-pearson classification. The Journal of Machine Learning Research, 14(1):3011–3040, 2013.
>
> [3] Xin Tong, Yang Feng, and Jingyi Jessica Li. “Neyman-Pearson Classification Algorithms and NP Receiver Operating Characteristics”.
>
> **The paper lacks a proper conclusion or discussion of future directions/limitations of the work which I feel would enhance the paper, especially is it is a very dense technical work.**
>
> Thanks for your suggestion. We will add a new section for the conclusion and future directions.
>
> **The experimental results do not illustrate a case where improper handling of the source data harms Type I error**
>
> Thanks for bringing up this question. We have added additional experiments in the appendix of the updated draft. In that experiment, the type-I error of the only-source method is very large, while the type-I error of the proposed method remains close to that of the only-target method, with only a slight deviation.
>
> We also noted in the appendix that, in the implementation, several constants appear in the quantities $\epsilon_{i,D}$ for $i \in \{S, T\}$. These $\epsilon$-terms arise from uniform concentration inequalities controlling the gap between empirical and population errors. In theory, such inequalities are conservative and their constants are not tight; in practice, however, these parameters can be tuned more carefully. Consequently, while the proposed method provides practical insight into designing an adaptive transfer-learning procedure, further tuning and additional experiments are needed to select these constants appropriately, as they directly influence the final error performance.
>
> **Could $g$ functions instead be introduced earlier before restrictive sets being defined?**
>
> The $g$-functions are introduced specifically for the parameterized setting—i.e., as functions of $\theta$—to illustrate the optimization procedure more transparently. However, when we define the restrictive sets (e.g.,$\hat{\mathcal{H}}'$, $\hat{\mathcal{H}}'_{1,D}$), we intentionally avoid assuming any particular parameterization of the hypotheses.
>
> **Typos**
>
> Thanks for pointing out the typos. We have updated the draft accordingly.
>
> **Question 1: Can this be extended to the case where some relationship between the target and source distributions are known and can be leveraged for example the density ratio or some other relationship?**
>
> Thank you for raising this question, which indeed suggests a promising future research direction. The optimal Neyman–Pearson classifier (i.e., the Bayes classifier) depends on the density ratio between class 1 and class 0. Estimating this classifier requires nonparametric techniques, as explored in Tong (2013) [1]. A natural extension is to study nonparametric transfer learning, where one aims to relate the source and target domains through their density ratios. Since the density ratio is the key quantity underlying the Bayes classifier, developing a nonparametric transfer learning framework that characterizes how these ratios can be estimated or transferred across domains would be a valuable direction for future research.
>
> [1] Xin Tong. A plug-in approach to neyman-pearson classification. JMLR, 2013.
>
> **Question 2:Are there illustrative examples of how the transfer modulus...**
>
> In the special case where $\mu_0=\mu_{0,S}=\mu_{0,T}$, our results recover those of [1]. In particular, Example1, where
> $\mu_0 = \mathcal{N}(-1,1)$, $\mu_{1,T} \sim \mathrm{Unif}[0,1]$, and
> $p_{1,S}(x) = \rho x^{\rho-1}\mathbf{1}_{\{x \in [0,1]\}}$, yields a corresponding transfer exponent equal to $\rho$, which can range from $1$ to $\infty$. Moreover, the transfer modulus functions introduced in the present paper imply bounds expressed in terms of the transfer exponent, as explained in the appendix.
>
> [1] M. M. Kalan, S. Kpotufe; Tight rates in supervised outlier transfer learning; ICLR-2024

---

> > ### Comment · Reviewer_dxrU · 2025-11-25
> >
> > Thank you for your response and additional experiments. From these I have a few remaining questions.
> >
> > # Additional Experiments
> > Regarding the experiment in Appendix D. You mention that constants in $\epsilon_{i,D}$ (there is a small typo there also, it should be $D\in\{S, T\}, i\in\{0,1\}$) could be causing the larger error however do you have intuition as to why the Type I error seems to diverge? Also, would it be possible to see results where $n_{0,T}$ grows to confirm that in this case the Type I error does converge to the target 0.1 level.
> >
> > # CP-Solver additional arguments
> > The $\epsilon$ and $\delta$ terms in Algorithm1 do not appear to be used within the body of the algorithm. Additionally, in Algorithm 2, $\epsilon$ is given as an argument to the function but $\delta$ is not. Can you explain the role these play?

---

> > > ### Author Response · Authors · 2025-11-26
> > >
> > > **There is a small typo there also, it should be $D\in{S, T}, i\in{0,1}$**
> > >
> > > Thanks for pointing out the typo. We have fixed it.
> > >
> > > **Do you have intuition as to why the Type I error seems to diverge?**
> > >
> > > Thanks for raising this question. When we increase the number of source samples, the curve does not diverge; rather, it increases and then stabilizes (see the updated plot in Figure 4 in the appendix, where the number of source samples increases up to 3000). The increase in type-I error from small to large sample sizes in the Gaussian example can be explained as follows. Consider the two cases of small and large numbers of samples. With a small number of samples, the empirical distribution is more likely to be heavy tailed, whereas with a large number of samples, the empirical source distribution for $\mu_{0,S}$ is close to the Gaussian population distribution. In the low-sample case, such a heavy-tailed empirical distribution yields a learned threshold that lies closer to the mean of the target distribution, leading to a smaller type-I error. In contrast, in the high-sample case, the empirical distribution aligns well with the source population distribution, which is far away from the target distribution; as a result, the threshold becomes stable and the curve flattens.
> > >
> > > **would it be possible to see results where $n_{0,T}$ grows to confirm that in this case the Type I error does converge to the target 0.1 level.**
> > >
> > > Please see Figure 5 in the appendix, where increasing the number of target samples shows that the Type-I error converges to the target level 0.1.
> > >
> > > **The $\epsilon$ and $\delta$ terms in Algorithm1 do not appear to be used within the body of the algorithm.**
> > >
> > > $\delta$ is the failure probability of the algorithm and $\epsilon$ is the accuracy of the output solution, which are indeed used for determining the total iteration number $N$, i.e., $N(\epsilon, \delta)$, and we make it explicit in the revised version. In Algorithm 2, in each calling of CP-Solver, we just need to set the failure probability to be $\frac{1}{5}\delta$ (which we make as an input parameter of CP-Solver in the revised version), so that the total probability is no larger than $\delta$. We have updated the Algorithm 1 and 2 accordingly in the revised version.

---

### Official Review · Reviewer_r8Y4 · 2025-10-30

**Soundness:** 2
**Presentation:** 2
**Contribution:** 2
**Rating:** 4
**Confidence:** 3

**Summary:**

Abstract: This paper investigates transfer learning within the Neyman–Pearson (NP) classification framework.

**Strengths:**

great theoretical results

**Weaknesses:**

**Weaknesses.**
The paper does not adequately discuss its limitations. In particular, the theoretical generalization analysis appears to rely on boundedness assumptions on the loss or surrogate loss, which is a common but restrictive condition.

Moreover, there are alternative transfer learning settings—such as fine-tuning and related approaches discussed in [1] and [2]—that are not compared against the proposed setup. Including such a comparison would clarify where the NP transfer setting stands relative to more standard transfer pipelines.

Finally, the motivation for studying the Neyman–Pearson formulation in the transfer learning context is not fully articulated. It would help to explain why NP constraints are especially relevant here, and to describe how competing approaches (e.g. $\alpha$-ERM or fine-tuning) would perform or be adapted in this setting.

[1] Aminian, Gholamali, Łukasz Szpruch, and Samuel N. Cohen. “Understanding Transfer Learning via Mean-field Analysis.” arXiv:2410.17128 (2024).
[2] Bu, Y., Aminian, G., Toni, L., Wornell, G. W., & Rodrigues, M. (2022). “Characterizing and understanding the generalization error of transfer learning with Gibbs algorithm.” AISTATS 2022, pp. 8673–8699.

**Questions:**

see weaknesses

---

> ### Author Response · Authors · 2025-11-22
>
> Thank you for your review and feedback.
>
> **The paper does not adequately discuss its limitations ... boundedness assumptions on the loss or surrogate loss, which is a common but restrictive condition.**
>
> We agree that the boundedness of the surrogate loss is an explicit assumption in our theoretical analysis. However, this assumption is standard and widely used in the machine learning literature, as it enables uniform convergence of empirical errors to their population counterparts through classical Rademacher complexity arguments.
>
> Importantly, this assumption is not restrictive in practice:
>
> 1) Many surrogate losses used in applications, including logistic and hinge losses, are clipped in implementations for numerical stability, which directly ensures boundedness.
>
> 2) Boundedness is also naturally satisfied when the hypothesis space is norm-bounded (for example, linear models or neural networks with bounded parameters), which is the standard setting in modern ML practice.
>
> 3) The models and losses used in our experiments already comply with these conditions, so the assumption does not limit the practical applicability of our method.
>
> We will add a discussion in the manuscript to clarify that this assumption is not restrictive in practice.
>
> **There are alternative transfer learning settings—such as fine-tuning and related approaches discussed in [1] and [2] ... how competing approaches (e.g. $\alpha$-ERM or fine-tuning) would perform or be adapted in this setting.**
>
> We will add a brief discussion clarifying that these mentioned methods are not directly related to our setting. In particular, $\alpha$-ERM assigns weights $\alpha$ and $1-\alpha$ to the target and source empirical losses, respectively, but it remains an unconstrained optimization problem and therefore does not control the Type-I error nor guarantee any bound on the Type-I error, which is essential in Neyman-Pearson classification. The same limitation applies to fine-tuning methods, which also optimize a single unconstrained loss, whereas the Neyman-Pearson problem requires minimizing the Type-II error subject to a Type-I constraint. We will include a short paragraph explaining this distinction in the revised manuscript.
>
> Furthermore, extending fine-tuning methods to the Neyman–Pearson setting, which is more challenging than vanilla classification since one must control two types of errors simultaneously, is nontrivial. Developing an adaptive fine-tuning approach that also avoids negative transfer would be a paper on its own.
>
> **The motivation for studying the Neyman–Pearson formulation in the transfer learning context is not fully articulated**
>
> In the first paragraph of the introduction, we provided real application examples from medicine, climate science, and cybersecurity, where controlling both types of errors is crucial, and where class imbalance makes classical loss-based methods insufficient for guaranteeing control of both errors. Nevertheless, we will include additional real-world examples where simultaneously controlling both Type-I and Type-II errors is critical.

---

> > ### Comment · Reviewer_r8Y4 · 2025-11-22
> > **feedback**
> >
> > Thanks for responses.
> >
> > I was wondering if authors want to revise draft for rebuttal.
> > You are allowed to revise submission for rebuttal.

---

> > > ### Author Response · Authors · 2025-11-22
> > >
> > > We have updated the manuscript. In the appendix, we added the discussion of the α-ERM and fine-tuning methods and cited the mentioned papers. Due to the 9-page limit, we plan to include this material in the main manuscript in the 10-page camera-ready version. Moreover, revising the structure of the paper for the suggested section would exceed the current page limit, and we will incorporate these changes in the final version. We have also included the additional experiments requested by other reviewers in the appendix.

---

> > > > ### Comment · Reviewer_r8Y4 · 2025-11-23
> > > >
> > > > Based on current guideline, you can provide 10 pages during rebuttal. please also highlight changes in blue color to help reviewers following changes.
> > > >
> > > > ''Paper length:
> > > > At the time of submission, the main text should be 9 pages or fewer. During the discussion/rebuttal phase and for the camera ready, the page limit will be increased to 10 pages to allow for new results/discussions. This limit will be strictly enforced. Papers with main text beyond the page limit will be desk-rejected.''

---

> > > > > ### Author Response · Authors · 2025-11-25
> > > > >
> > > > > We have addressed the comments in the updated manuscript, with the changes highlighted in blue.

---

> > > > > > ### Comment · Reviewer_r8Y4 · 2025-11-26
> > > > > >
> > > > > > Thanks. I revised my score accordingly. I gave score 8. Not 10 due to limited experiments and limited applications of results.

---

### Official Review · Reviewer_oLLY · 2025-11-01

**Soundness:** 3
**Presentation:** 2
**Contribution:** 2
**Rating:** 6
**Confidence:** 2

**Summary:**

The authors study Neyman-Pearson classification in the context of transfer learning, where both source and target domains provide positive and negative samples. The objective is to minimize the false negative (Type-II) rate while ensuring the false positive (Type-I) rate remains below a specified threshold on the target distribution. While prior work has mainly focused on the case where the negative distributions are identical between source and target, this paper explores the more general scenario where they differ. The primary contribution is an adaptive error bound on the target false negative rate, which interpolates between using only target samples and leveraging additional source samples. The authors further propose a computational algorithm with matching error guarantees, assuming convexity for both the loss function and classifier class. Experimental results validate the algorithm's adaptive control of the Type-II error while adhering to the target Type-I error constraint.

**Strengths:**

This paper addresses Neyman-Pearson classification within the transfer learning framework, which is highly relevant to the conference. The Neyman-Pearson classification setup is practical due to its importance in high-stakes real-world applications, and transfer learning is motivated by the scarcity of labeled samples in the target domain.

The authors present an algorithm with error bound guarantees that avoids negative transfer, even when both the negative and positive source and target distributions differ. As previous works are limited to the setting where the negative source and target distributions are identical, this represents a theoretical contribution to a more general scenario.

The resulting error bound is characterized by a newly proposed quantity assessing the transferability of source samples, namely, the transfer modulus. This measure has the potential to capture non-polynomial associations between the source and target distributions, whereas previous transfer exponents are restricted to polynomial relationships.

The authors also provide a computational algorithm achieving the same error bound guarantee. Even under convexity assumptions, they demonstrate a polynomial runtime algorithm for transfer Neyman-Pearson classification.

**Weaknesses:**

The practical scenarios in which $R_{\varphi,\mu_1,T}(h^\ast_{S,T,\alpha}) - R_{\varphi,\mu_1,T}(h^\ast_{T,\alpha})$ is small are unclear. If this term is not small, using only target samples dominates the rate. In such cases, the benefit of transfer learning is not adequately demonstrated. Since $h^\ast_{S,T,\alpha}$ is defined as the maximizer of $R_{\varphi,\mu_1,S}$, the excess error between $h^\ast_{S,T,\alpha}$ and $h^\ast_{T,\alpha}$ could remain large even when the source and target distributions are similar. The justification for using $h^\ast_{S,T,\alpha}$ as the pivot in the error bound should be explained more clearly.

The experimental results suggest that false positive rate control is not adaptive. Theorem 1 asserts that the additive error in false positive rate control is adaptive, meaning it is bounded by the minimum of the error rates for using only target samples and the transfer method. However, Figure 3 shows that the false positive rate error is comparable to that of the method using only target samples. This discrepancy between theory and experiment should be addressed.

The paper's structure could be improved. For example, the setup section (Section 3) combines the problem formulation of Neyman-Pearson classification under the transfer learning framework, discussion of prior results, and challenges in constructing the proposed algorithm. A clearer separation of these components would be beneficial.

**Questions:**

- In what concrete scenarios is the excess term $R_{\varphi,\mu_1,T}(h^\ast_{S,T,\alpha}) - R_{\varphi,\mu_1,T}(h^\ast_{T,\alpha})$ provably small? Can you characterize such cases via properties of the source/target distributions?
- Why is $h^\ast_{S,T,\alpha}$ the right pivot for the analysis? Could similar or tighter bounds be derived by pivoting around alternatives such as a target‑only optimizer on a mixed/importance‑reweighted distribution, and how would that affect the rates?
- Theorem 1 suggests adaptive control of the false positive rate (Type‑I), yet Figure 3 appears comparable to the target‑only method. What explains this discrepancy, and under what conditions should we expect adaptivity to be visible empirically?

---

> ### Author Response · Authors · 2025-11-22
>
> Thank you for your review and feedback.
>
> **The practical scenarios in which $R_{\varphi,\mu_1,T}(h^\ast_{S,T,\alpha}) - R_{\varphi,\mu_1,T}(h^\ast_{T,\alpha})$ is small are unclear.**
>
> Thanks for bringing up this question. The justification for using $h^\ast_{S,T,\alpha}$ as the pivot is discussed in Remark 1, and we clarify it further here. First note that, in the extreme case where the source and target are identical, we have $\mu_{1,S} = \mu_{1,T}$ and $\mathcal{H}_S(\alpha) = \mathcal{H}_T(\alpha)$.
>
> This implies that $h^\ast_{S,T,\alpha} = h^\ast_{T,\alpha}$ and therefore
>
> $R_{\varphi,\mu_1,T}(h^\ast_{S,T,\alpha}) - R_{\varphi,\mu_1,T}(h^\ast_{T,\alpha}) = 0$.
>
> When the deviation between the source and target is small, the corresponding difference departs only slightly from zero and remains small. This behavior is illustrated in Figure 1, which reflects a common practical setting where the distributions are Gaussian. In the illustrated example, the deviations vary continuously as the source moves away from the target.
>
> Furthermore, when $\mu_{0,S} = \mu_{0,T}$ but $\mu_{1,S}$ may differ arbitrarily from $\mu_{1,T}$. In this situation, $h^\ast_{S,T,\alpha} = h^\ast_{S,\alpha}$ , which recovers the result in [1], where it is shown to be minimax optimal. This holds for any distributions.
>
> Furthermore, this is a natural choice: when the amount of target data is very limited, the natural approach is to minimize the source class-1 error while ensuring that the type-I error, evaluated on both the source and the target domains, remains below $\alpha$. These cases, together with the practical Gaussian examples, justify the choice of the pivot. However, establishing that it is the optimal choice in all settings and scenarios remains an interesting direction for future research, particularly for proving the minimax optimality of the derived bound. We will clarify this further in the manuscript.
>
> [1] M. M. Kalan, S. Kpotufe; Tight rates in supervised outlier transfer learning; ICLR-2024
>
> **The experimental results suggest that false positive rate control is not adaptive...This discrepancy between theory and experiment should be addressed.**
>
> We clarify that the concern arises from a misinterpretation of Theorem 1, and there is no discrepancy between Theorem 1 and the experimental results shown in Figure 3. Theorem 1 does not state that, if one trains two functions separately on the source and target using the same pre-threshold $\alpha$, then the proposed transfer-learning procedure will achieve, on the target domain, a type-I error less than the minimum of the type-I errors of those two functions. Instead, the type-I error bound in Theorem 1 depends on the quantity $\alpha_S$ defined in Section 4, which in some cases may be substantially larger than $\alpha$.
>
> Theorem 1 guarantees that the type-I error of the proposed method is bounded above by the minimum of the following two quantities:
>
> (i) the type-I error of the classifier trained only on the target data, and
>
> (ii) $\phi_0(\max\\{\alpha_S + \epsilon_{0,S},\alpha+\epsilon_{0,S}\\})$,
>
> while the type-I error of the source-only method is bounded by $\phi_0(\alpha+\epsilon_{0,S})$. In Figure 2, the type-I error is lower than that of the target-only method and close to that of the source-only method, which corresponds to the case where $\alpha$ is larger, or at least not much smaller, than $\alpha_S$. In Figure 3, the type-I error is close to that of the target-only method and far from that of the source-only method, reflecting the scenario where $\alpha$ is much smaller than $\alpha_S$.
>
> Furthermore, in Figure 1, we have illustrated the cases $\alpha_S > \alpha$ and $\alpha_S < \alpha$ using Gaussian examples to clarify this behavior.
>
> Thank you for raising this question, and we will clarify it further in the manuscript.
>
> **The paper’s structure could be improved.**
>
> Thanks for your suggestion. We will further improve the structure of the paper and incorporate your suggestions.
>
> **Questions:**
>
> We have addressed the questions in the paragraphs above.

---

### Official Review · Reviewer_Wj1g · 2025-11-01

**Soundness:** 3
**Presentation:** 3
**Contribution:** 3
**Rating:** 6
**Confidence:** 3

**Summary:**

The paper studies transfer learning for Neyman–Pearson (NP) classification when both class-conditional distributions may shift between source and target domains. It proposes a two-stage adaptive procedure: (i) calibrate a source-side Type-I threshold `α̂_S` so that the source constraint aligns with the target NP constraint, thereby pruning hypotheses that would violate the target Type-I bound; (ii) within this restricted set, leverage source class-1 data (and target data) to further reduce Type-II risk. The paper derives generalization bounds stated via a *transfer modulus*—functions `ϕ₀` and `ϕ₁` translating source performance to target—and shows recovery of prior results when `μ₀,S = μ₀,T`. It also provides a computational oracle via a sequence of convex programs (SGDA-style) with gradient-complexity guarantees, and evaluates the method on two climate datasets where locations define domain shifts. :contentReference[oaicite:0]{index=0}

**Strengths:**

I was not previously familiar with the NP classification setting, but I find both the problem and the authors’ extensions interesting. The authors also provide sufficient theoretical guarantees for their proposed algorithm, along with supportive simulation studies. In terms of novelty and substance, I believe the paper is worthy of publication at ICLR, although this is not my primary area of expertise.

**Weaknesses:**

1. It is somewhat difficult to discern a strong novelty in the transfer-learning extension, even though the authors present solid theoretical results and simulations.

**Questions:**

1. The theory relies on a convex hypothesis class $H$ and convex, Lipschitz, bounded surrogate losses, whereas the experiments use an MLP. Do MLPs constitute a convex hypothesis class?
2. Can we derive similar theoretical guarantees for transfer learning under an overlapping-support assumption between the source and target datasets?

---

> ### Author Response · Authors · 2025-11-21
>
> Thank you for your review and feedback.
>
> **It is somewhat difficult to discern a strong novelty in the transfer-learning extension:**
>
> A common challenge in many machine learning applications is data scarcity. To mitigate this, one can leverage additional datasets that are similar but not identical. However, determining how to exploit an auxiliary dataset effectively is itself challenging: if the auxiliary data are not sufficiently related, performance may even deteriorate. It is therefore essential to develop an adaptive method for exploiting such datasets; one that benefits from them when they are informative while remaining robust to negative transfer when they are not. Even in vanilla classification, achieving adaptive transfer is challenging; in Neyman–Pearson classification it becomes even more difficult, as one must control two types of errors simultaneously.
>
> In this paper, we develop an adaptive transfer-learning approach that, for the first time, provides theoretical guarantees on both statistical performance and computational efficiency in a fully general setting where distribution shifts may occur in both class-conditional distributions.
>
> **Q1: The theory relies on a convex hypothesis class ... whereas the experiments use an MLP. Do MLPs constitute a convex hypothesis class?**
>
> MLPs are not, in general, a convex hypothesis class. However, as explained in Section 5 of the manuscript, one can consider the convex hull of an MLP class, and doing so preserves the Rademacher complexity. Since our theorem holds for all data distributions, working with a convex class ensures that small deviations in the type-I error do not lead to large deviations in the type-II error, thereby avoiding pathological cases that are unlikely to arise in real datasets.
>
> **Q2: Can we derive similar theoretical guarantees for transfer learning under an overlapping-support assumption:**
>
> Since our theorem holds for all data distributions, any restricted class of source and target distributions is automatically covered by our theorem. However, imposing additional structure or prior knowledge on the relationship between the source and target distributions may allow one to derive tighter bounds.

---

### Meta-Review · Area_Chair_mTko · 2026-01-10

**Summary:**

This paper addresses transfer learning in the framework of Neyman–Pearson classification. The Authors consider a scenario in which both class-conditional distributions may shift between the source and target domains. They derive an adaptive procedure to solve this problem. The paper also establishes generalization bounds and shows how prior results are recovered as a special case. Empirical results on two climate datasets are presented.

The Reviewers are generally positive about the paper and raise mostly minor issues, such as how the theory relates to practical scenarios, potential mismatches between the theoretical and empirical results, the relation to alternative transfer learning settings, and the motivation behind the considered setup. These concerns have been successfully addressed by the Authors in the rebuttal.

**Reviewer Concerns:**

The concerns raised in reviews have been successfully addressed by the Authors in the rebuttal.

**Reviewer Scores:**

- Reviewer Wj1g would likely keep the score of 6 or slightly increase it
- Reviewer oLLY would likely keep the score of 6 or slightly increase it
- Reviewer r8Y4 would likely increase the score to 8 as written in their comment
- Reviewer dxrU would likely keep the score of 8

---

### Decision · Program_Chairs · 2026-01-26

Accept (Poster)